# Successor-Predecessor Intrinsic Exploration

**Changmin Yu**[1,2] **Neil Burgess**[1,*] **Maneesh Sahani**[2,*] **Samuel J. Gershman**[3,*]
[1]Institute of Cognitive Neuroscience; [2]Gatsby Computational Neuroscience Unit;
UCL, London, United Kingdom
[3]Department of Psychology, Harvard University, Cambridge, United States
[*] Joint senior authors

## Abstract

Exploration is essential in reinforcement learning, particularly in environments where external rewards are sparse. Here we focus on exploration with *intrinsic rewards*, where the agent transiently augments the external rewards with self-generated intrinsic rewards. Although the study of intrinsic rewards has a long history, existing methods focus on composing the intrinsic reward based on measures of future prospects of states, ignoring the information contained in the retrospective structure of transition sequences. Here we argue that the agent can utilise retrospective information to generate explorative behaviour with structure-awareness, facilitating efficient exploration based on global instead of local information. We propose *Successor-Predecessor Intrinsic Exploration* (SPIE), an exploration algorithm based on a novel intrinsic reward combining prospective and retrospective information. We show that SPIE yields more efficient and ethologically plausible exploratory behaviour in environments with sparse rewards and bottleneck states than competing methods. We also implement SPIE in deep reinforcement learning agents, and show that the resulting agent achieves stronger empirical performance than existing methods on sparse-reward Atari games.

## 1 Introduction

The study of exploration in reinforcement learning (RL) has produced a broad range of methods [1, 2], ranging from simple methods such as pure randomization [3, 1, 4], to more sophisticated methods such as targeted exploration towards states with high uncertainty [5–7] and implicit exploration with entropy maximization [8, 9]. Intrinsic exploration, a highly effective class of methods, uses intrinsic rewards based on the agent's current knowledge of the environment, hence informing targeted exploration towards states with high predictive uncertainty or state occupancy diversity [10, 11, 6, 12]. However, existing approaches define the intrinsic reward based solely on prospective or empirical marginal information about future states, ignoring retrospective information (e.g., does a given state always precedes the goal state, hence should be more frequently traversed?). We argue that the retrospective information contains useful signals about the connectivity structure of the environment, hence could facilitate more efficient targeted exploration. For example, consider a clustered environment with bottleneck states connecting the clusters (Figure 1a), exploration based on local information (e.g., visitation counts) would discourage the agent from traversing bottleneck states, despite the key roles these states play in connecting different clusters. Guiding the agents to visit such "bottleneck" states in the face of minimal local information gain is essential in driving efficient and biologically plausible exploration. Here we study the contribution of retrospective information for global exploration with intrinsic motivation.

One of the most successful recent intrinsic exploration algorithms [12] uses the successor representation (SR; [13, 14]) to generate intrinsic rewards. The SR represents each state in terms of successor states. The row norms of the SR can be used as an intrinsic reward that generalises count-based

37th Conference on Neural Information Processing Systems (NeurIPS 2023).

exploration [12]. As we discuss in Section 3, the SR contains not only prospective information, but also retrospective information about expected predecessors. This information can be utilised to construct a novel intrinsic reward which overcomes some of the problems associated with purely prospective intrinsic rewards, such as untargeted exploration, the augmented reward function is non-stationary, and asymptotic uniformity.

We provide a brief overview of background and relevant literature in Section 2, and formally introduce the novel intrinsic exploration method, *Successor-Predecessor Intrinsic Exploration* (SPIE), in Section 3. We propose two instantiations of SPIE for discrete and continuous state spaces, with comprehensive empirical examinations of properties of SPIE in discrete state space. We show that SPIE facilitates more efficient exploration, in terms of improved sample efficiency of learning and higher asymptotic return, through empirical evaluations on both discrete and continuous environments in Section 4.

## 2 Background and related work

**Reinforcement Learning Preliminaries**. We consider the standard RL problem in Markov Decision Processes (MDP), defined by the tuple, $\langle \mathcal{S}, \mathcal{A}, \mathcal{P}, \mathcal{P}_0, \mathcal{R}, \gamma \rangle$, where $\mathcal{S}$ is the state space, $\mathcal{A}$ is the action space, $\mathcal{P} : \mathcal{S} \times \mathcal{A} \to \Delta(\mathcal{S})$ is the state transition distribution (where $\Delta(\mathcal{S})$ is the probability simplex over $\mathcal{S}$), $\mathcal{P}_0 \in \Delta(\mathcal{S})$ is the initial state distribution, $\mathcal{R} : \mathcal{S} \times \mathcal{A} \to \mathbb{R}$ is the reward function, and $\gamma \in (0, 1)$ is the discount factor. The goal for an RL agent is to learn the optimal policy that maximises value (expected cumulative discounted reward): $\pi^*(a|s) = \mathrm{argmax}_\pi\, q^\pi(s, a), \forall (s, a) \in \mathcal{S} \times \mathcal{A}$, where $\pi : \mathcal{S} \to \Delta(\mathcal{A})$ is the policy, and $q^\pi(s, a)$ is the state-action value function:

$$q^\pi(s, a) = \mathbb{E}_{\mathcal{P}^\pi}\left[\textstyle\sum_{\tau=0}^{\infty} \gamma^\tau \mathcal{R}(s_\tau, a_\tau)|s_0 = s, a_0 = a\right] = \mathbb{E}_{\mathcal{P}^\pi}[\mathcal{R}(s, a) + \gamma q^\pi(s', a')]\,, \tag{1}$$

where $\mathcal{P}^\pi(s'|s) = \sum_a \pi(a|s)\mathcal{P}(s'|s, a)$ is the marginal state transition distribution given $\pi$[1]. The second equality is the recursive form of the action value function known as the *Bellman equation* [15], which underlies temporal difference learning [1]:

$$\hat{q}^\pi(s_t, a_t) \leftarrow \hat{q}^\pi(s_t, a_t) + \alpha\delta_t, \quad \delta_t = r_t + \gamma\hat{q}^\pi(s_{t+1}, a_{t+1}) - \hat{q}^\pi(s_t, a_t)\,, \tag{2}$$

where $\hat{q}^\pi(s_t, a_t)$ is the current estimate of the action values (with respect to $\pi$), $\delta_t$ is the (one-step) temporal difference (TD) error. We will study the effect of different intrinsic rewards on the performance of online TD learning (SARSA) in discrete state MDPs.

**The successor representation**. The SR is defined as the expected cumulative discounted future state occupancy under the policy[2]:

$$\mathbf{M}[s, s'] = \mathbb{E}_{\mathcal{P}^\pi}\left[\textstyle\sum_{\tau=0}^{\infty} \gamma^\tau \mathbb{1}(s_\tau, s')|s_0 = s\right] = \mathbb{E}_{\mathcal{P}^\pi}\left[\mathbb{1}(s_0, s') + \gamma\mathbf{M}(s_1, s')|s_0 = s\right]. \tag{3}$$

Given the recursive formulation, it is possible to learn the SR matrix online with TD learning. Given the transition tuple, $(s_t, a_t, r_t, s_{t+1}, a_{t+1})$, the update is

$$\hat{\mathbf{M}}[s_t, s'] \leftarrow \hat{\mathbf{M}}[s_t, s'] + \alpha\delta_t^{\mathbf{M}}, \quad \delta_t^{\mathbf{M}} = \mathbb{1}(s_t, s') + \gamma\hat{\mathbf{M}}[s_{t+1}, s'] - \hat{\mathbf{M}}[s_t, s']\,, \tag{4}$$

Note that these equations are analogous to TD learning for value function estimation, except that in this case the function being learned is a vector-valued (one-hot) representation of future states.

**First-occupancy representation**. The SR captures the expected cumulative discounted state occupancy over all future steps. However, in many real-world and simulated tasks, it may be preferable to reach the goal state as quickly as possible instead of as frequently as possible. In this spirit, Moskovitz et al. [16] introduced the *First-occupancy Representation* (FR). Formally, the FR matrix in a discrete MDP is defined by

$$\begin{aligned}\mathbf{F}[s, s'] &= \mathbb{E}_{\mathcal{P}^\pi}\left[\sum_{\tau=0}^{\infty} \gamma^\tau \mathbb{1}(s_\tau = s', s' \notin \{s_{0:\tau}\})|s_0 = s\right] \\ &= \mathbb{E}_{\mathcal{P}^\pi}\left[\mathbb{1}(s_t, s') + \gamma(1 - \mathbb{1}(s_t, s'))\mathbf{F}[s_{t+1}, s']|s_t = s\right]\,,\end{aligned} \tag{5}$$

[1]Note that unless otherwise stated, we assume deterministic MDP, i.e., $\mathcal{P}(s'|s, a)$ is a delta function.

[2]For notational simplicity, we keep the policy dependence implicit. Similar notational choice holds for all quantities discussed in the rest of the paper ($\mathbf{F}$ and $\mathbf{N}$).

where $\{s_{0:\tau}\} = \{s_0, s_1, \ldots, s_{\tau-1}\}$. The recursive formulation implies that there is an efficient TD learning rule for online learning of the FR matrix. Given the transition tuple $(s_t, a_t, r_t, s_{t+1}, a_{t+1})$, the update rule is

$$\hat{\mathbf{F}}[s_t, s'] \leftarrow \hat{\mathbf{F}}[s_t, s'] + \alpha \delta_t^{\mathbf{F}}, \quad \delta_t^{\mathbf{F}} = \mathbb{1}(s_t, s') + \gamma(1 - \mathbb{1}(s_t, s'))\hat{\mathbf{F}}[s_{t+1}, s'] - \hat{\mathbf{F}}[s_t, s'] , \tag{6}$$

**Intrinsic exploration in RL**. Here we focus on exploration with intrinsic motivation, where the agent augments the external rewards with self-constructed intrinsic rewards based on its current knowledge of the environment.

$$r^{\text{tot}}(s, a) = r^{\text{ext}}(s, a) + \beta r^{\text{int}}(s, a) , \tag{7}$$

where $r^{\text{ext}}(s, a)$ denotes the extrinsic environmental reward, $r^{\text{int}}(s, a)$ denotes the (possibly non-stationary) intrinsic reward, and $\beta$ is a multiplicative scaling factor controlling the relative balance of $r^{\text{ext}}(s, a)$ and $r^{\text{int}}(s, a)$. The intrinsic reward often operates by motivating the agent to move into under-explored parts of the state space in the absence of extrinsic reinforcement. Many types of intrinsic rewards have been proposed, including functions of state visitation counts [17–19], predictive uncertainty of value estimation [5], and predictive error of forward models [10, 6, 20]. In a closely related work, Zhang et al. [21] proposes NovelD, which constructs the episode-specific non-negative intrinsic reward based on the difference between the novelty measures of temporally adjacent states along a trajectory. However in contrast to SPIE (discussed later), the key difference is that NovelD does not explicitly utilise the retrospective information for exploration and the associated intrinsic reward is episode-dependent.

## 3 Successor-Predecessor Intrinsic Exploration

Existing intrinsic exploration methods construct intrinsic rewards based on either the predictive information in a temporally forward fashion (e.g., predictive error), or the empirical marginal distribution (e.g., count-based exploration). Here we argue that the retrospective information inherent in experienced trajectories, though having been largely overlooked in the literature, could also be utilised as a useful exploratory signal. Specifically, consider the environment in Figure 1a (*Cluster-simple*), where the discrete grid world is separated into two clusters connected by a "bottleneck" state. Whenever the starting and reward locations are in different clusters, the bottleneck state, $s_*$, always precedes the goal state, regardless of the trajectory taken. Hence, the frequent predecessor state (e.g., $s_*$), to the goal state should be traversed despite the fact that immediate information gain by traversing the state is minimal. In the absence of extrinsic reward, if only utilising learned prospective information based on past experience (e.g., the norm of the online-learned SR [12]), the intrinsic motivation for exploration is merely local hence would discourage transitions into bottleneck states. However, the retrospective information can be utilised to identify the state transitions that connect different sub-regions of the state space, hence incorporating the connectivity information of the state space into guiding exploration, allowing the agent to escape local exploration and navigate towards bottleneck states to reach distant regions.

We develop *Successor-Predecessor Intrinsic Exploration* (SPIE) algorithm utilising intrinsic rewards based on both prospective and retrospective information from past trajectories. Below we provide instantiations of SPIE based on the SR for discrete and continuous state spaces.

**SPIE in discrete state space**. We define the *SR-Relative* (SR-R) intrinsic reward, which is defined as the SR of the future state from the current state minus the sum of the SRs of the future state from all states. Formally, given a transition tuple, $(s, a, r, s', a')$, we define the SR-R intrinsic reward as:

$$r_{\text{SR-R}}(s, a) = \hat{\mathbf{M}}[s, s'] - ||\hat{\mathbf{M}}[:, s']||_1 = - \sum_{\tilde{s} \in \mathcal{S}, \tilde{s} \neq s} \hat{\mathbf{M}}[\tilde{s}, s'] , \tag{8}$$

The above equation holds in deterministic MDPs (i.e., when $s'$ is a function of $(s, a)$). We note that the $j$-th column of the SR matrix represents the expected discounted occupancy to state $j$, starting from every state, hence constituting a *temporally backward* measure of the accessibility of state $j$ [22]. Therefore, $r_{\text{SR-R}}(s, a)$ consists of both a prospective measure ($\hat{\mathbf{M}}[s, s']$) and a retrospective measure ($||\hat{\mathbf{M}}[:, s']||_1$), and exploring with $r_{\text{SR}}$ is an instantiation of SPIE in discrete MDPs. Intuitively, $r_{\text{SR-R}}(s)$ can be interpreted as penalising transitions leading into states $s'$ that are frequently reached from many states other than $s$, hence providing an intrinsic motivation for guiding the agent towards states that are harder to reach in general, e.g, boundary states and bottleneck states. We thoroughly

investigate the individual contribution of prospective and retrospective information through ablation studies in Appendix B.4, and we observe that prospective information alone does not yield optimal exploration performance, whereas utilising only the retrospective information does not degrade exploration efficiency, indicating the importance of global topological information contained in the retrospective information for intrinsic exploration.

In a closely related work, Machado et al. [12] showed that $r_{\text{SR}}(s) = 1/||\hat{\mathbf{M}}[s,:]||_1$ can be used as an intrinsic reward that facilitates exploration in sparse reward settings. They additionally showed that the row norm of the online-learned SR matrix implicitly approximates the state visitation counts, so the resulting behaviour resembles count-based exploration. However, a key issue associated with $r_{\text{SR}}$ is that the asymptotic exploratory behaviour is uniformly random across all states, i.e., $||\mathbf{M}[s,:]||_1 \rightarrow 1/(1-\gamma), \forall s \in \mathcal{S}$. We note that exploration involves learning of both the environmental transition structure $\mathcal{P}^{\pi}$ and the reward structure $\mathcal{R}$. Hence, were the SR matrix to be known *a priori* (hence $\mathcal{P}^{\pi}$ could be implicitly derived), no intrinsic motivation would be introduced at any state and the resulting agent regresses back to random exploration, omitting further efficient exploration for learning $\mathcal{R}$. Since $r_{\text{SR-R}}$ contains the sum of columns of the SR matrix, the asymptotic uniformity in $r_{\text{SR}}$ no longer holds, yielding non-trivial intrinsic exploration even when the SR matrix is known and fixed *a priori*, allowing continual exploration for learning the reward structure despite sparse extrinsic reinforcement.

**Analysis of $r_{\text{SR-R}}$ with pure exploration in grid-worlds**. We examine exploration based on $r_{\text{SR-R}}(s)$ in discrete grid-worlds with different topologies (Fig. 1a). We first consider pure exploration in the absence of extrinsic reward, and evaluate the exploratory behaviours of $4$ RL agents with different intrinsic rewards, in terms of their state coverage. The agents we consider are: vanilla SARSA [1]; SARSA with $r_{\text{SR}}$ (SARSA-SR; [12]); SARSA with $r_{\text{FR}}(s) = ||F[s,:]||_1$ (SARSA-FR; [16]); and SARSA with $r_{\text{SR-R}}$ (SARSA-SRR); the pseudocode for SARSA-SRR can be found in Appendix). We consider $4$ different grid-world environments with different configurations (Figure 1a), namely, $10 \times 10$ open-field grid (*OF-small*); $10 \times 10$ grid with two rooms (*Cluster-simple*); $10 \times 10$ grid with $4$ rooms (*Cluster-hard*); and $20 \times 20$ open-field grid (*OF-large*).

Exploration efficiency is quantified as the number of timesteps taken to cover $50\%$, $90\%$ and $99\%$ of the state space. The value estimates for all states are initialised to be $0$. Due to the absence of extrinsic reward, the vanilla SARSA agent is equivalent to a random walk policy, which acts as a natural baseline. We observe from Figure 1b that SARSA-SRR yields the fastest coverage of the state space amongst all considered agents. The SARSA-FR agent yields similar state coverage efficiency as SARSA. SARSA-SR performs poorly in all 4 grid-worlds, failing to achieve $50\%$ state coverage within $8000$ timesteps in all environments other than the simplest one (*OF-small*). Moreover, we observe that SARSA-SRR performed consistently across the $4$ considered grid configurations, whereas all other agents experienced significant degradation in exploration efficiency as the size and complexity of the environments increase.

We note that in addition to improved exploration efficiency, SARSA-SRR exhibits "cycling" behaviour in pure exploration in the $20 \times 20$ two-cluster environment (Figure 6e), spending the majority of its time exploring in one cluster and periodically traverses the "bottleneck" states to explore the opposing clusters upon sufficient coverage of the current cluster. Such "cycling" strategy exhibits short-term memory of recent states and consistent long-term planned exploration towards regions more distant in history. This is potentially advantageous for environments with non-stationary reward structures ([23]), such as real-world foraging, which require continual exploration for identifying new rewards. We verify the capability of SARSA-SRR for dealing with non-stationary reward structure in Section 4 (Figure 3).

The complexity of analysing the properties of SARSA-SRR is two-fold: the online learning of the SR matrix and the online update of the Q-values. By assuming the SR matrix is known and fixed throughout training,[3] we observe from Figure 1c that SARSA-SRR consistently outperforms all competing methods, similar to what we observed when the SR (FR) matrix is learned online. Additionally, we observe that the exploration efficiency for all three intrinsic exploration agents drops when using the intrinsic reward constructed with the fixed SR (FR), but SARSA-SRR yields minimal decrease comparing to the significant degradation with SARSA-SR and SARSA-FR. Hence, we have empirically confirmed that the improved exploration efficiency does not stem solely from the online learning of the SR matrix, but is a property of $r_{\text{SR-R}}$. Another long-standing issue with many existing

[3]We assume the policy the fixed SR matrix is dependent on is the random walk policy unless otherwise stated.

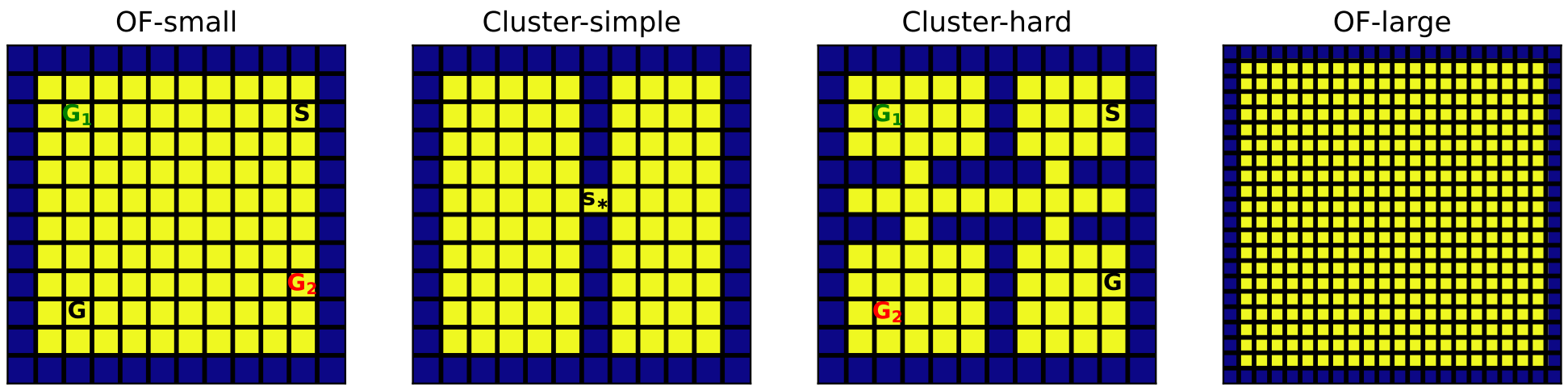

(a)

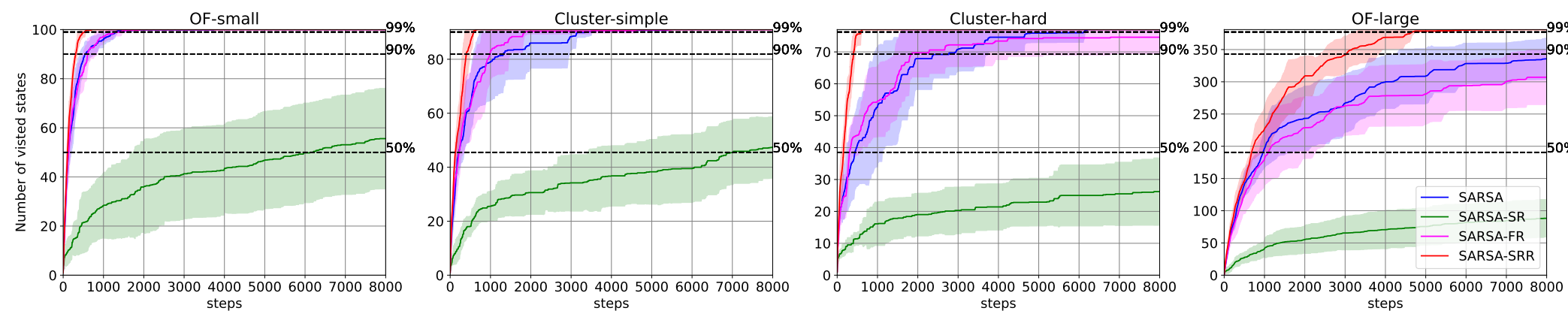

(b)

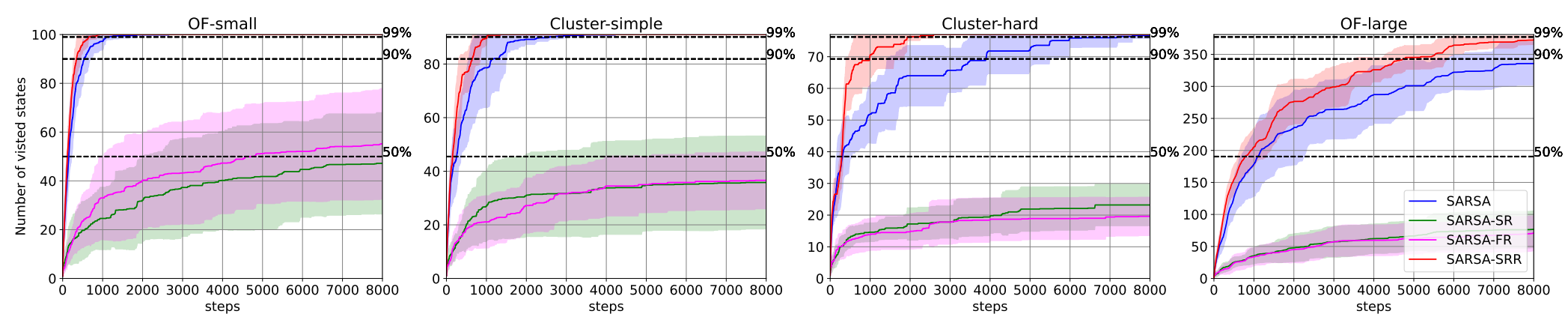

(c)

Figure 1: **Evaluation of exploration efficiency in grid worlds.** (a) Grid worlds with varying size and complexity. 'S' and 'G' in *OF-small* and *Cluster-hard* represents the start and goal states in the goal-oriented reinforcement learning task; colored $G_1$ and $G_2$ in *OF-small* and *Cluster-hard* represent the changed goal locations (see the non-stationary reward experiment in Section 4), $s_*$ in *Cluster-simple* denote the bottleneck state. (b-c) Accumulated number of states visited against exploration timesteps, for all considered agents in all grid-worlds in with (a) online-learned SR matrix (b) and fixed SR matrix (c). All reported results are averaged over 10 random seeds (shaded area denotes mean $\pm$ 1 standard error). Hyperparameters can be found in Appendix.

intrinsic exploration methods is the non-stationary nature of the associated intrinsic bonus. By fixing the SR (FR) matrix, the associated $r_{\text{SR-R}}$ is stationary whilst still yielding high exploration efficiency, hence validating the utility of SPIE.

**SPIE in continuous state space with deep RL**. In order to generalise $r_{\text{SR-R}}$ to continuous state space, we replace the SR with successor features (SF; [24]).

$$\boldsymbol{\psi}^\pi(s,a) = \mathbb{E}\left[\sum_{k=0}^{\infty} \gamma^k \boldsymbol{\phi}_{t+k} | s_t = s, a_t = a\right] = \boldsymbol{\phi}(s_{t+1}) + \mathbb{E}\left[\boldsymbol{\psi}^\pi(s_{t+1}, \pi(s_{t+1})) | s_t = s, a_t = a\right] \tag{9}$$

where $\boldsymbol{\phi}(s,a)$ is a feature representation such that $r(s,a) = \boldsymbol{\phi}(s,a) \cdot \mathbf{w}$, with weight parameter $\mathbf{w}$. The recursive formulation for SF admits gradient-based learning of $\phi$ by minimising the following squared TD loss.

$$\delta_t^{\text{SF}} = \mathbb{E}\left[\left(\phi(s_t, a_t) + \gamma\psi(s_{t+1}, a_{t+1}) - \psi(s_t, a_t)\right)^2\right], \tag{10}$$

where the transition tuple $(s_t, a_t, s_{t+1}, a_{t+1})$ can be taken from either online samples (SARSA-like) or sampled from offline trajectories (Q-learning-like). We previously noted that the column of the SR matrix provides a marginal retrospective accessibility of states, facilitating stronger exploration. However, there is no SF-analogue of the column of the SR matrix. We therefore construct the retrospective exploration objective with the *Predecessor Representation* (PR), which was proposed to measure how often a given state is preceded by any other state given the expected cumulative discounted preceding occupancy [25]. The formal definition for the PR matrix under discrete MDP,

$\mathbf{N} \in \mathbb{R}^{|\mathcal{S}|\times|\mathcal{S}|}$, is defined as following.

$$\mathbf{N}[s, s'] = \mathbb{E}_{\tilde{\mathcal{P}}^\pi}\left[\sum_{\tau=0}^{n} \gamma^\tau \mathbb{1}(s, s_{n-\tau}) | s_n = s'\right] = \mathbb{E}_{\tilde{\mathcal{P}}^\pi}\left[\mathbb{1}(s, s_n) + \gamma \mathbf{N}[s, s_{n-1}]\right], \tag{11}$$

where the expectation is based on $\tilde{\mathcal{P}}^\pi(s_t = s | s_{t+1} = s') = \frac{\mathcal{P}^\pi(s,s')z(s)}{z(s')}$, the *retrospective* transition model, and $z(s) = \lim_{t\to\infty} \mathbb{E}_{\mathcal{P}^\pi}[\mathbb{1}(s_t = s)]$, denotes the stationary distribution given policy $\pi$.

Utilising the recursive formulation for the PR matrix, we can again derive a TD-learning rule. Namely, given the transition tuple, $(s_t, a_t, r_t, s_{t+1}, a_{t+1})$, we have the following update rule.

$$\hat{\mathbf{N}}'[\tilde{s}, s_{t+1}] = \hat{\mathbf{N}}[\tilde{s}, s_{t+1}] + \alpha \delta_t^{\mathbf{N}}, \quad \delta_t^{\mathbf{N}} = \mathbb{1}(s_{t+1}, \tilde{s}) + \gamma \hat{\mathbf{N}}[\tilde{s}, s_t] - \hat{\mathbf{N}}[\tilde{s}, s_{t+1}]. \tag{12}$$

The SR and PR have a reciprocal relationship (proof in appendix):

$$\mathbf{N}\text{diag}(\mathbf{z}) = \text{diag}(\mathbf{z})\mathbf{M}, \tag{13}$$

where $\text{diag}(\mathbf{z}) \in \mathbb{R}^{|\mathcal{S}|\times|\mathcal{S}|}$ denotes the diagonal matrix whose diagonal elements corresponds to the discrete stationary distribution of the MDP under the current policy.

Similar to how SF generalises SR, we propose the "Predecessor Feature" (PF) that generalises PR.

$$\boldsymbol{\xi}^\pi(s) = \mathbb{E}\left[\sum_{k=0}^{\infty} \gamma^k \mu_{t-k} | s_{t+1} = s\right] = \boldsymbol{\mu}(s_{t+1}) + \gamma \mathbb{E}\left[\boldsymbol{\xi}^\pi(s_t) | s_{t+1} = s, a_t = a\right]. \tag{14}$$

Similarly to the SF, the recursive definition of the PF again allows a simple expression of the TD error for gradient-based learning of the PF.

$$\delta_t^{\text{PF}} = \mathbb{E}\left[(\phi(s_{t+1}) + \gamma \xi(s_t) - \xi(s_{t+1}))\right], \tag{15}$$

We utilise the norms of SF and PF to replace the row sums in discrete settings for tractable approximation to $r_{\text{SR-R}}$ in continuous state spaces. We use the same feature vector, $\phi$, for computing the SF and PF. In order to ensure the SF and PF are of similar scales across the state space, we normalise $\phi(s)$ such that $||\phi(s)||_2 = 1$ for all $s$. Contrary to how we define $r_{\text{SR-R}}$ as the difference between the SR and the column sum of the SR in discrete MDPs[4], we find that setting the intrinsic reward as the difference between the reciprocal of the norms of the SF and the PF yields better empirical performance. We hence define the continuous Successor-Predecessor intrinsic reward as follows.

$$r_{\text{SF-PF}} = \frac{1}{||\boldsymbol{\phi}(s_{t+1})||_1} - \frac{1}{||\boldsymbol{\psi}(s_t, a_t)||_1} \tag{16}$$

**Details of deep RL implementation of $r_{\text{SF-PF}}$.** We instantiate $r_{\text{SF-PF}}$ based on a Deep Q Network (DQN; [26]), with auxiliary predictive reconstruction task ($\mathcal{L}_{\text{recon}} = \mathbb{E}\left[(s_{t+1} - \hat{s}_{t+1})^2 | s_t\right]$, where $\hat{s}_{t+1}$ is the predicted next state), and separate heads for computing the q-values, the SF, and the PF, respectively (Figure 2). We call this model DQN-SF-PF. Note that, following Machado et al. [12], the intermediate feature representation $\phi$ is trained given only the predictive reconstruction and value learning supervisions, and not updated given the TD error in the learning of the SF or the PF (the filled black circle in Figure 2 indicating the `stop_gradient` operation). We adopt the same set of hyperparameters and architecture for the DQN as reported in Oh et al. [27]. To make the comparison consistent, we utilise the

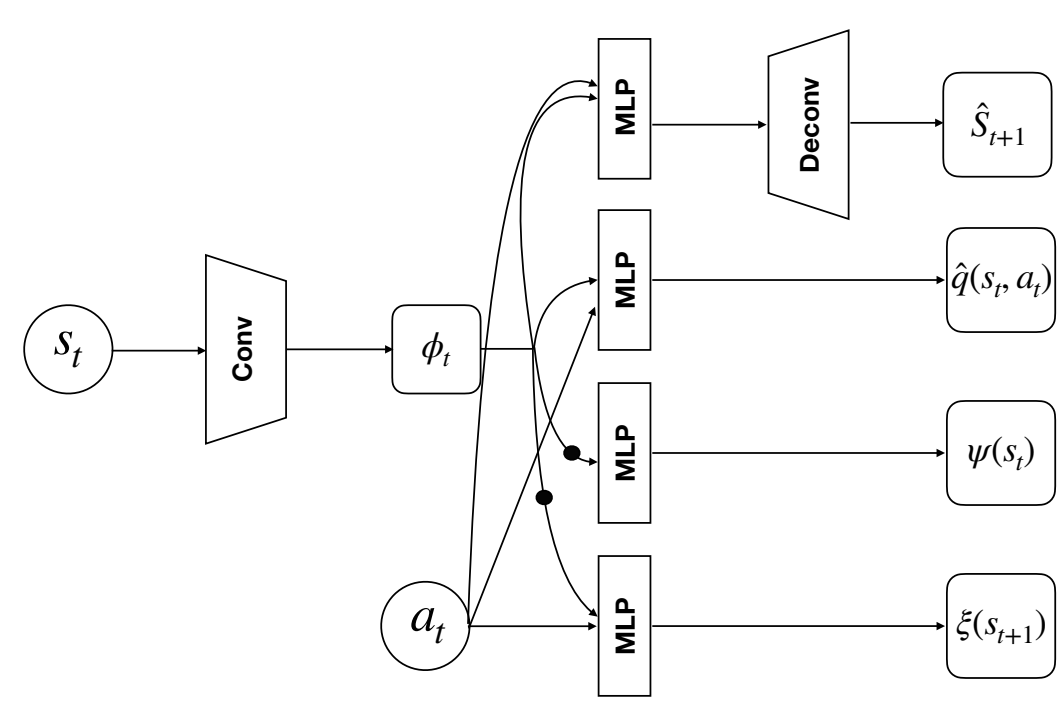

Figure 2: Graphical illustration of the neural network architecture of DQN-SF-PF for Atari games. Note that the state feature vector is L2-normalised, $\phi(s) = \frac{\tilde{\phi}(s)}{||\tilde{\phi}(s)||_2}$, where $\tilde{\phi}(s)$ is the raw output of the convolutional encoder.

[4]Note we refer to discrete/continuous MDP as an MDP with discrete/continuous state and action space.

Table 1: Evaluations SARSA-SRR and related baseline agents on RiverSwim and SixArms (averaged over 100 seeds, numbers in the parentheses represents standard errors).

| | SARSA | SARSA-SR | SARSA-FR | SARSA-SRR | SARSA-SR-PR |
|---|---|---|---|---|---|
| RiverSwim | 25,075 | 1,197,075 | 1,547,243 | $\mathbf{2,547,156}$ | $\mathbf{2,857,324}$ |
| | (1,224) | (36,999) | (34,050) | (479,655) | (419,922) |
| SixArms | 376,655 | 1,025,750 | 119,149 | $\mathbf{2,199,291}$ | $\mathbf{1,845,229}$ |
| | (8,449) | (49,095) | (42,942) | (1,024,726) | (1,032,050) |

mixed Monte-Carlo return loss [28, 12], defined as following.

$$\mathcal{L}_q = \mathbb{E}\left[\left((1-\tau)\delta_{\text{TD}}(s,a) + \tau\delta_{\text{MC}}(s,a)\right)^2\right] ,$$
$$\text{where } \delta_{\text{MC}}(s,a) = \sum_{t=0}^{\infty} \gamma^t \left(r(s_t,a_t) + \beta r_{\text{SF-PF}}(s_t,a_t;\theta^-)\right) - q(s,a;\theta) , \tag{17}$$

where $\delta_{\text{TD}}$ denotes the standard TD error for q-values (Eq. 2), $\tau$ is the scaling factor controlling the contribution of the TD error and the Monte Carlo error, and $\theta$ and $\theta^-$ denote the parameters for the online and target DQN-SF-PF, respectively. Hence the overall loss objective for training DQN-SF-PF is as following.

$$\mathcal{L}_{\text{DQN-SF-PF}} = w_{\text{q}}\mathcal{L}_q + w_{\text{SF}}\delta^{\text{SF}} + w_{\text{PF}}\delta^{\text{PF}} + w_{\text{recon}}\mathcal{L}_{\text{recon}} , \tag{18}$$

where $w_{\text{q/SF/PF/recon}}$ denotes the scaling factors for the respective loss terms. The complete set of hyperparameters for DQN-SF-PF can be found in the Appendix.

## 4 Experiments

**Classical hard exploration tasks**. We evaluate performance of the discrete SPIE agent (and other considered agents in Section 3: SARSA, SARSA-SR, SARSA-FR) on two classical hard exploration tasks commonly studied in the PAC-MDP literature, *RiverSwim* and *SixArms* [29] (appendix Figure 5). In both tasks, environment transition dynamics induce a bias towards states with low rewards, leaving high rewards in states that are harder to reach. Evaluation of the agents is based on the cumulative reward collected within $5000$ training steps.

We observe from Table 1 that SARSA-SRR significantly outperforms all other considered agents. Moreover, in order to further justify the utility of $\mathcal{R}_{\text{SR-R}}$ in driving exploration, we run ablation studies by evaluating the performance of variants of SARSA-SRR (Appendix B.4). Ablation studies reveal the importance of combining both prospective and retrospective information for exploration, as well as the benefits of dynamic balancing exploring uncertain states and bottleneck states.

In order to validate the replacement of column norm of the SR with column norm of the PR in the construction of $r_{\text{SR-R}}$, given the reciprocal relationship (Eq. 13), we empirically evaluate the performance of the SARSA agent with the alternative intrinsic reward, $r_{\text{SR-PR}}(s,a) = \hat{\mathbf{M}}[s,s'] - ||\hat{\mathbf{N}}[:,s']||_1$. SARSA-SR-PR yields comparable performance as SARSA-SRR on both RiverSwim and SixArms (Table 1), empirically justifying the instantiation of SPIE with the PR for capturing the retrospective information.

**Goal-oriented / sparse-reward tasks**. We next evaluate the agents on grid world tasks with a single terminal goal state (Figure 1a; *OF-small* and *Cluster-hard*). All non-terminal transitions yield rewards of $-1$, and transitions into the goal state generates a reward of $0$. Such goal-directed or sparse-reward tasks require efficient exploration. We examine both open-field and clustered grid-worlds. In *OF-small* and *Cluster-hard* tasks, SARSA-SRR outperforms both vanilla SARSA and SARSA-SR in terms of sample efficiency (Figure 3). In addition, SARSA-SRR yields more stable training and performance is more robust across different random seeds. Note that the navigation performance of SARSA-SR during training is highly unstable, which might attribute to its equivalence to count-based exploration given that visitation count is only a local measure for exploration. Somewhat surprisingly, the improvement for SARSA-SRR is more significant in open-field grid world (*OF-small*) rather than

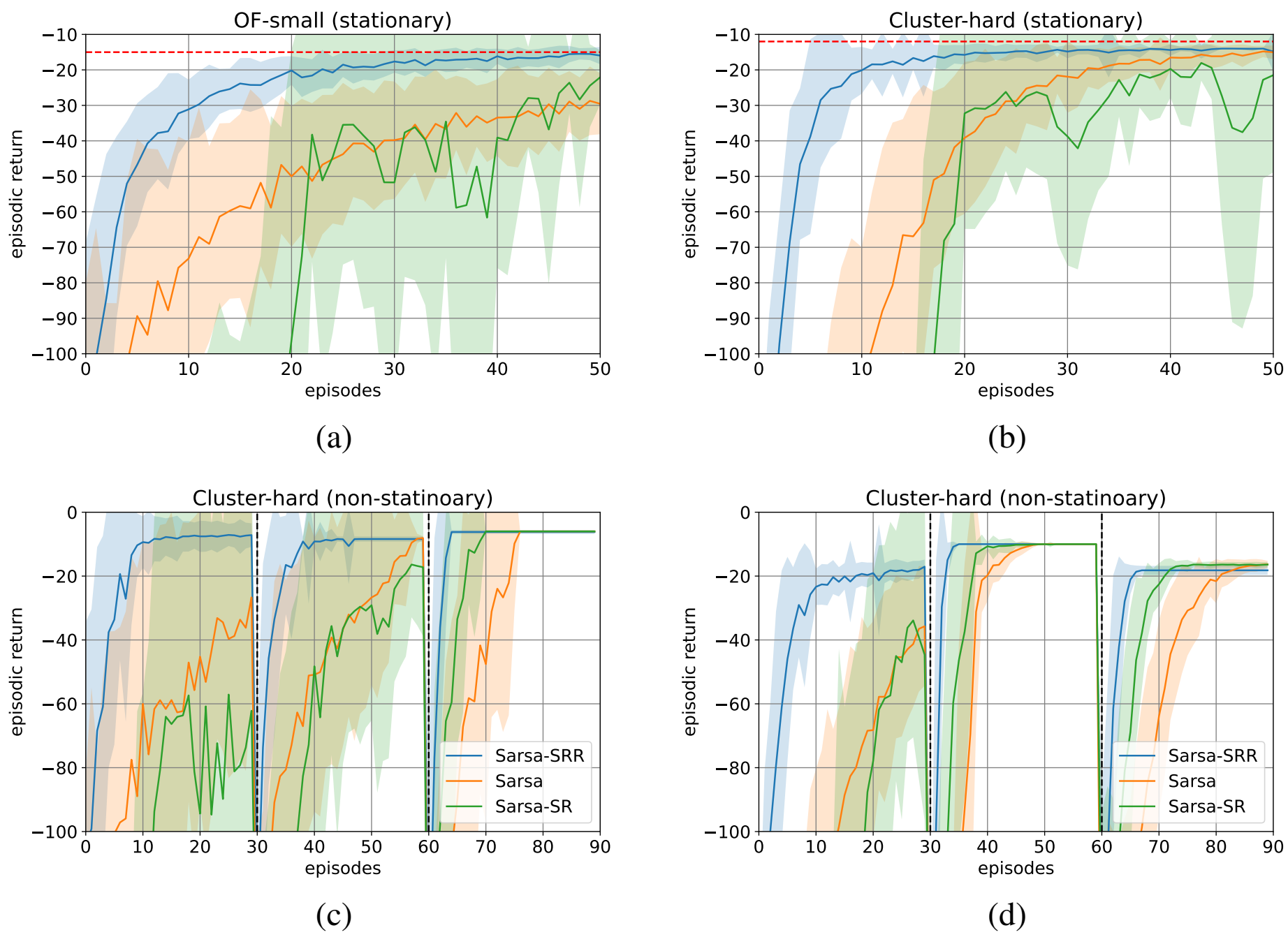

Figure 3: **Goal-oriented navigation in grid worlds.** Evaluations of SARSA, SARSA-SR and SARSA-SRR on *OF-small* (a) and *Cluster-hard* (b) grid worlds (Figure 1a) with stationary reward structure, and on *OF-small* (c) and *Cluster-hard* (d) with *non-stationary* reward structures. The red dashed horizontal line represents the shorted path distance. The black dashed vertical lines represent the time point at which the goal change occurs.

the clustered grid world (*Cluster-hard*), in contrast to the pure exploration experiments (Figure 1b). Nevertheless, the improvement is strong and consistent.

In many real-world tasks, the environment is inherently dynamic, requiring continual exploration for adapting the optimal policy with respect to the non-stationary task structure. One such example is random foraging, where foods are depleted upon consumption, and new rewards appear in new locations. As argued in Section 3, SARSA-SRR yields "cycling" exploratory behaviour (Figure 6), hence could facilitate continual exploration that is potentially suitable for such non-stationary environments. To empirically justify the hypothesis, we consider the Non-Markovian Reward Decision Process (NMRDP; [23]), where the reward changes dynamically given the visited state sequence. We instantiate the NMRDPs in the grid worlds, *OF-small* and *Cluster-hard*, where there are three reward states ($G$, $G_1$, $G_2$; Figure 1a) that are sequentially activated (and deactivated) every 30 episodes. As shown in Figure 3c and 3d, we observe that SARSA-SRR consistently outperforms SARSA and SARSA-SR, reaching the new goal states in increasingly shorter timescales. This supports our idea that SPIE provides a more ethologically plausible exploration strategy for dealing with non-stationarity. However, we note that the main focus of the current paper is on improved exploration within a single task, instead of over a stream of inter-related tasks. Here we provide preliminary evidence of potential applicability of SPIE in such continual exploration setting, and we leave more rigorous investigation in this direction for future work.

**Linear function approximation for continuous state spaces**. We next evaluate SPIE with function approximation. As a first step, we consider the linear features before moving onto the deep RL setting. We consider the *MountainCar* task (Figure 4a; [30]), with sparse reward structure, where we set the reward to 0 for all transitions into non-terminal states (the terminal state is indicated by the flag on the top of the right hill). We utilise Q-learning with linear function approximation, where we define the linear features to be the 128-dimensional random Fourier features (RFF [31]; Figure 4b). The SF and the PF are defined given the RFF, and are learned via standard TD-learning (Eq. 10; 15). The performance (over the first 1000 training episodes) of the resulting linear-Q agents with $r_{\text{SF}}$ and

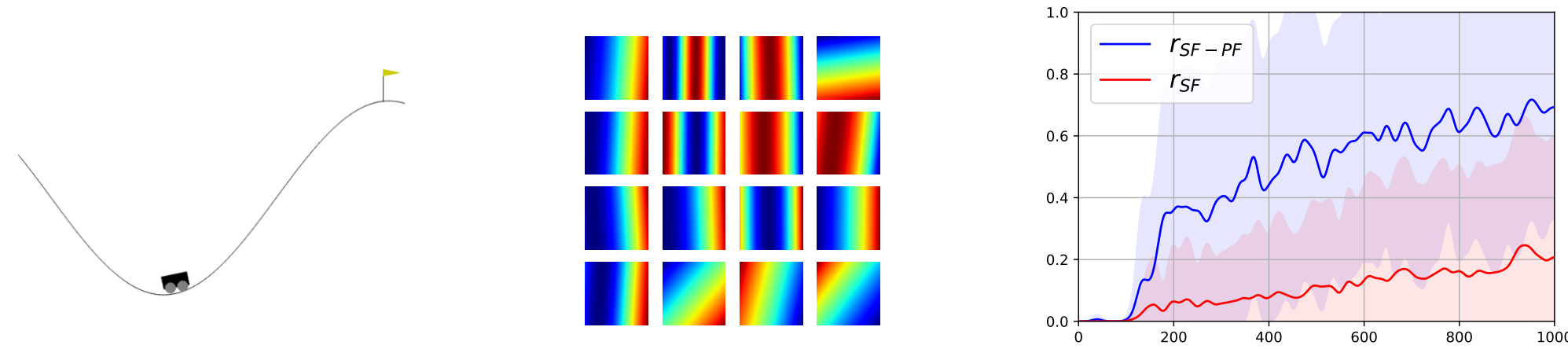

(a) (b) (c)

Figure 4: **Evaluation of SPIE with linear features in MountainCar.** (a) Graphical demonstration of MountainCar environment; (b); Example random Fourier features; (c) Evaluations of Q-learning with linear function approximation with intrinsic rewards $r_{\text{SF}}$ and $r_{\text{SF-PF}}$ on MountainCar. Reported results are averaged over 10 random seeds.

Table 2: Evaluations of SPIE with deep RL implementation on hard-exploration Atari games (averaged over 10 random seeds, numbers in the parentheses are 1 standard errors).

| | DQN | DQN$^{\text{MMC}}$ | RND | DQN$^{\text{MMC}}$-SR | DQN$^{\text{MMC}}$-SF-PF |
|---|---|---|---|---|---|
| Freeway | 32.4 (0.3) | 29.5 (0.1) | 28.2 (0.2) | 29.4 (0.1) | 27.5 (0.2) |
| Gravitar | 118.5 (22.0) | 1078.3 (254.1) | 714.1 (105.9) | 457.4 (120.3) | 1223.0 (408.9) |
| Mont. Rev. | 0.0 (0.0) | 0.0 (0.0) | 528 (314.0) | 1395.4 (1121.8) | 1530.0 (1072.1) |
| Private Eye | 1447.4 (2,567.9) | 113.4 (42.3) | 61.3 (53.7) | 104.4 (50.4) | 488.2 (390.9) |
| Solaris | 783.4 (55.3) | 2132.6 (394.8) | 1395.2 (401.7) | 1890.1 (163.1) | 2455.8 (262.0) |
| Venture | 4.4 (5.4) | 1220.1 (51.0) | 953.7 (167.3) | 1348.5 (56.5) | 1274.0 (133.2) |

$r_{\text{SF-PF}}$ is shown in Figure 4c. The agent with $r_{\text{SF-PF}}$ outperforms the opposing agent significantly, empirically justifying the utility of SPIE in the linear function approximation regime.

**Deep RL instantiation of SPIE in Atari games**. We empirically evaluate DQN$_{\text{SF-PF}}$ on 6 Atari games with sparse reward structures [32]: Freeway, Gravitar, Montezuma's Revenge, Private Eye, Solaris, and Venture. We follow the evaluation protocol as stated in Machado et al. [33], where we report the averaged evaluation scores over 10 random seeds given $10^8$ training steps. The agent takes (stacked) raw pixel observations as inputs. Across all 4 games, the $\beta$ values are set to $0.07$ and the discounting factor $\gamma = 0.995$. We adopt the epsilon-annealing scheme as in [28], which linearly decreases $\epsilon$ from 1.0 to 0.1 over the first $10^6$ frames. We train the network with RMSprop, with standard hyperparameters, learning rate $0.00025$, $\epsilon = 0.001$ and decay equals $0.95$ [26]. The discounting factors for value learning and online learning of the SF and the PF are set to $0.99$. The scaling factors in Eq. 18 are set such that the different losses are on roughly similar scales: $w_q = 1$, $w_{\text{SF}} = 1500$, $w_{\text{PF}} = 1500$, $w_{\text{recon}} = 0.001$. More implementation details can be found in Appendix.

We compare DQN-SF-PF with vanilla DQN trained with standard TD error, vanilla DQN trained with the MMC loss ($\mathcal{L}_q$), Random Network Distillation (RND; [20]), DQN-SR trained with the MMC loss [12] (Table 2). All agents are trained with the predictive reconstruction auxiliary task. By comparing with our main baseline, DQN-SR, we observe that DQN-SF-PF significantly outperforms DQN-SR on Four games (Gravitar, Montezuma's Revenge, Private Eye and Solaris), whilst yielding similar performance on the remaining two games (Freeway and Venture). Moreover, DQN-SF-PF outperforms RND, a state-of-the-art Deep RL algorithm for exploration, on all 6 games. The empirical difference is not only reflected in the asymptotic performance, but also in the sample efficiency of learning. Specifically, for Montezuma's Revenge, one of the hardest exploration games in the Atari suite, our agent achieves near asymptotic performance (defined as the score given $10^8$ training steps) with only $\sim 8 \times 10^6$ training frames, whereas the performance of DQN-SR saturates at $\sim 2.4 \times 10^7$ training frames (with a lower score). We emphasise that the main aim of our empirical evaluations is to validate the utility of SPIE exploration objective as a simple modification to DQN. In principle, SPIE can be integrated with any state-of-the-art RL agent, and different instantiations of SPIE could be implemented to deal with the task at hand. We leave such investigation for future work.

## 5 Conclusion

The development of more efficient exploration algorithms is essential for practical implementation of RL agents in real-world environment where sample efficiency and optimality are vital to success. Here, we propose a general intrinsically motivated exploration framework, SPIE, where we construct intrinsic rewards by combining both prospective and retrospective information contained in past trajectories. The retrospective component provides information about the connectivity structure of the environment, facilitating more efficient targeted exploration between sub-regions of state space given structure awareness (e.g., robust identification of the bottleneck states; Figure 1a). SPIE yields more sample efficient exploration in discrete MDPs under complete absence of external reinforcement. Moreover, a side benefit we observe empirically is that SPIE exhibits ethologically plausible exploratory behaviour during exploration in grid worlds (i.e., cycling between different clusters of states). In continuous state space, we developed a novel generalization of the predecessor representation, the predecessor features, for capturing retrospective information in continuous spaces. Empirical evaluations on both discrete and continuous MDPs demonstrate that SPIE yields improvements over existing intrinsic exploration methods, in terms of sample efficiency of learning and asymptotic performance, and for adapting to non-stationary reward structures.

We instantiate SPIE using the SR and the PR, but we note that SPIE is a general framework that can be implemented with other formulations (e.g., predictive error in a temporally backward direction [34, 35]) and with more advanced neural architectures (including those currently unthought of). Although here we have examined the empirical properties of SPIE, the theoretical underpinnings for SPIE and the bottleneck seeking exploratory behavior bears further investigation. Specifically, more work needs to be done to probe the theoretical property of using SF and PF in continuous settings. Our definition of $r_{\text{SR-R}}$ overlaps with the successor contingency [25, 36], which has long been recognised for learning causal relationship between predictors and reward [37]. An interesting venue for future work is to investigate the implications of SPIE for causally guided exploration in RL. Another interesting direction for future work is to investigate the implications of SPIE in human exploration, where we could utilise SPIE to investigate how human balance local (e.g., visitation counts) versus global (e.g., environment structure) information for exploration in sequential decision tasks [38, 39].

## Acknowledgement

We thank Franziska Brändle, James Heald, and Ted Moskovitz for useful discussions, and anonymous reviewers for valuable comments. This work is funded by the UKRI, DeepMind, the Gatsby Charitable Foundation, the Simons Foundation, the Wellcome Trust, and the Harvard Brain Initiative and by the Center for Brains, Minds and Machines (CBMM).

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

# A More Details on Predecessor Representation

Here we provide proofs of the reciprocal relationship between the SR and the PR.

**Proposition A.1.** $\mathbf{N}$*diag*$(\mathbf{z})$ = *diag*$(\mathbf{z})\mathbf{M}$*, where diag*$(\mathbf{z})$ *is the diagonal matrix with the diagonal elements as the vector* $\mathbf{z}$*, and* $\mathbf{z}$ *is the vector of stationary distribution of* $\mathcal{P}^\pi$ *(i.e.,* $\mathbf{z}[i] = \lim_{t\to\infty} \mathbb{E}_{\mathcal{P}^\pi}[s_t = i]$.

*Proof.* Given the formal definition of the SR and the PR (Eq. 3; 11), we have the following analytical expressions.

$$\mathbf{M} = (\mathbf{I} - \gamma\mathcal{P}^\pi)^{-1}; \quad \mathbf{N} = (\mathbf{I} - \gamma\tilde{\mathcal{P}}^\pi)^{-1}; \tag{19}$$

where $\tilde{\mathcal{P}}^\pi$ is the temporally reversed transition distribution. Assume matrix formulation of $\mathcal{P}^\pi$ and $\tilde{\mathcal{P}}^\pi$, $\mathbf{P}$ and $\tilde{\mathbf{P}}$ in $\mathbb{R}^{|\mathcal{S}|\times|\mathcal{S}|}$, we have the following.

$$\begin{aligned} \tilde{\mathbf{P}}_{ij} &= \mathbb{P}(s_t = i | s_{t+1} = j) = \frac{\mathbb{P}(s_{t+1} = j | s_t = i)\mathbb{P}(s_t = i)}{\mathbb{P}(s_{t+1} = j} = \frac{\mathbf{P}_{ij}\mathbf{z}_i}{\mathbf{z}_j}\,, \\ \Rightarrow& \tilde{\mathbf{P}}\mathrm{diag}(\mathbf{z}) = \mathrm{diag}(\mathbf{z})\mathbf{P}\,, \end{aligned} \tag{20}$$

Substituting the reciprocal relationship between $\tilde{\mathbf{P}}$ and $\mathbf{P}$ into the definition of the PR, we have the following.

$$\begin{aligned} \mathbf{N} &= \left(\mathbf{I} - \gamma\mathrm{diag}(\mathbf{z})\mathbf{P}\mathrm{diag}(\mathbf{z})^{-1}\right)^{-1}\,, \\ \mathbf{N}\mathrm{diag}(\mathbf{z}) &= \left(\mathbf{I} - \gamma\mathrm{diag}(\mathbf{z})\mathbf{P}\mathrm{diag}(\mathbf{z})^{-1}\right)^{-1}\mathrm{diag}(\mathbf{z}) \\ &= \left(\mathrm{diag}(\mathbf{z})^{-1}(\mathbf{I} - \gamma\mathrm{diag}(\mathbf{z})\mathbf{P}\mathrm{diag}(\mathbf{z})^{-1})\right)^{-1} \\ &= \left((\mathbf{I} - \gamma\mathbf{P})\mathrm{diag}(\mathbf{z})^{-1}\right)^{-1} \\ &= \mathrm{diag}(\mathbf{z})\left((\mathbf{I} - \gamma\mathbf{P})\right)^{-1} \\ &= \mathrm{diag}(\mathbf{z})\mathbf{M} \end{aligned} \tag{21}$$

□

# B Further results on tabular hard exploration tasks.

## B.1 Graphical illustration of tabular hard-exploration tasks.

The demos of RiverSwim and SixArms is shown in Figure 5. In both tasks, the environmental transition dynamics impose asymmetry, biasing the agent towards low-rewarding states that are easier to reach, with greater rewards available in hard-to-reach states.

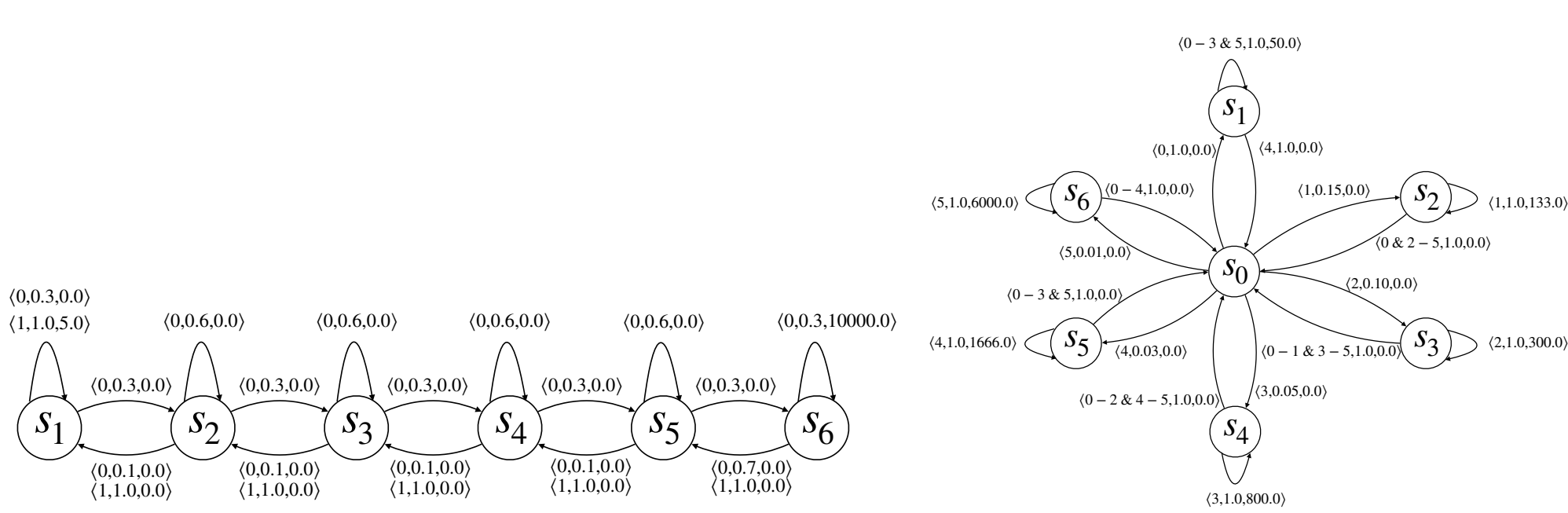

(a) RiverSwim (b) SixArms

Figure 5: Discrete MDPs. Transition probabilities are denoted by ⟨action, probability, reward⟩. In RiverSwim (a), the agent starts in state 1 or 2. In SixArms (b), the agent starts in state 0.

Table 3: Evaluations on RiverSwim and SixArms with intrinsic rewards based on fixed SR/FR (averaged over 100 seeds, numbers in the parentheses represents standard errors).

| | SARSA-SR | SARSA-FR | SARSA-SRR |
|---|---|---|---|
| RiverSwim | 327,402 | 278,096 | **3,096,913** |
| | (787,118) | (666,752) | (230,059) |
| SixArms | 969,781 | 1,143,037 | **2,059,424** |
| | (2,895,306) | (1,939,021) | (3,292,936) |

### B.2 Pseudocode for SARSA-SRR.

We provide the pseudocode for SARSA-SRR in Algorithm 1. We note that SARSA, SARSA-SR and SARSA-FR utilise the similar algorithm, but only replacing the intrinsic bonus.

**Algorithm 1** Pseudocode for SARSA-SRR

```
Require: α, η, γ, γ_SR, β, ϵ
  s = env.reset();
  M = 0 ∈ R^{|S|×|S|};                                        ▷ Initialise the SR matrix as zero matrix
  Q = 0 ∈ R^{|S|×|A|};
  while not done do
      θ ∼ U(0, 1);
      if θ < ϵ then                                                              ▷ ϵ-greedy policy
          a ∼ U(A);
      else
          a = argmax_{a∈A} Q[s, a];
      end if
      s′, r, done = env.step(a);
      M[s, :] = M[s, :] + η (1(s) + γ_SR(1 − done)M[s′, :] − M[s, :]);   ▷ TD-learning of the SR
      r = r + β(M[s, s′] − ||M[:, s′]||_1);                          ▷ Constructing intrinsic reward
      θ′ ∼ U(0, 1);
      if θ′ < ϵ then
          a′ ∼ U(A);
      else
          a′ = argmax_{a∈A} Q[s′, a];
      end if
      Q[s, a] = Q[s, a] + α (r + γ(1 − done)Q[s′, a′] − Q[s, a]);
      s = s′;
  end while
```

### B.3 Evaluations given the fixed SR.

Conforming to our analysis of $r_{\text{SR-R}}$ with fixed SR (Section 3), we additionally evaluate SARSA-SR/FR/SRR with the corresponding intrinsic rewards constructed based on fixed SR/FR matrix on RiverSwim and SixArms (Table 3. Similar to what we found in the grid worlds (Figure 1c), both SARSA-SR and SARSA-FR perform worse than their online-SR counterparts (note one exception being SARSA-FR on SixArms). However, in contrast to the decrease in exploration efficiency of SARSA-SRR in grid worlds, we found that fixing the SR actually improves the performance of SARSA-SRR. Hence, in accord with our analysis in Section 3, the cause for the improved empirical performance of $r_{\text{SR-R}}$ does not lie solely in the online learning process of SR, but might stems from the inherent "bottleneck-seeking" property of $r_{\text{SR-R}}$.

### B.4 Ablation studies of SPIE in discrete tasks

We perform ablation studies on SARSA-SRR for further demonstration of the utility of the SPIE objective of combining both the prospective and retrospective information. We firstly show that prospective

Table 4: Ablation studies of SARSA-SRR on RiverSwim and SixArms.

| | SARSA-SRR | SARSA-SRR(a) | SARSA-SRR(b) | SARSA-SRR(c) |
|---|---|---|---|---|
| RiverSwim | $\mathbf{2,547,156}$ | 127,703 | $\mathbf{2,629,947}$ | 95,691 |
| | (479,655) | (530,564) | (930,170) | (181,216) |
| SixArms | $\mathbf{2,199,291}$ | 893,530 | $\mathbf{1,902,553}$ | 562,346 |
| | (1,024,726) | (2,601,324) | (2,211,960) | (1,748,455) |

information alone cannot yield strong exploration, whereas utilising solely the retrospective information maintains the strong explorative performance. We consider two variants of SARSA-SRR, SARSA-SRR(a) and SARSA-SRR(b), with the respective intrinsic rewards as following.

$$\mathcal{R}_{\text{SR-R(a)}}(s,a,s') = \hat{M}[s,s'] \,, \quad \mathcal{R}_{\text{SR-R(b)}}(s,a,s') = -||\hat{M}[:,s']||_1 \,, \tag{22}$$

From Table 4, we observe that utilising the prospective information alone for exploration yields suboptimal performance, hence empirically justifying the utility of the SPIE framework. However, we do observe that utilising the retrospective information alone yields near- or supra-optimal performance. Together, the results indicate that the global topological information contained in the retrospective information is essential for intrinsic exploration purposes.

We argue that the dynamic balancing between exploring states with high uncertainty and bottleneck states is a key factor driving the empirical success of SPIE. In order to test this hypothesis, we devise a variant of the $\mathcal{R}_{\text{SR-R}}$.

$$\mathcal{R}_{\text{SR-R(c)}} = ||\hat{M}[s,:]||_1 - \hat{M}[s,s'] \,, \tag{23}$$

Intuitively, $\mathcal{R}_{\text{SR-R(c)}}$ provides an intrinsic motivation for taking transitions that lead to states that are less reachable from $s$, which only yields exploration towards states of high uncertainty, but does not provide any motivation towards bottleneck states. Indeed, as we observe from Table 4 that SARSA-SRR(c) also yields suboptimal performance, providing empirical evidence supporting the benefits of SPIE in driving the agents towards bottleneck states.

# C Further results on exploration in grid worlds

## C.1 Transient dynamics of exploration.

We look more closely at the transient dynamics of the considered agents during pure exploration in *Cluster-simple-large* (where *Cluster-simple-large* denotes the $20 \times 20$ grid world with two clusters). We observe that in the absence of external reinforcement, SARSA-SR, regardless of based on intrinsic rewards given either online-learned or fixed SR matrix, exhibits minimal exploration (Figure 6a 6b). This is largely due to its local exploration behaviour. For SARSA-FR, we observe significant difference between using online-trained and fixed FR matrix, where exploration with intrinsic rewards based on fixed FR completed disrupts exploration, only exploring a small proportion of the environment. In contrast, we observe that SARSA-SRR consistently fully explores both clusters (repeatedly) under both conditions. Additionally, by closely examining the transient dynamics during the exploration phase, we observe the "cycling" behaviour[5].

## C.2 Effect of optimistic initialisation.

We note that across all considered SARSA agents, the Q values were initialised to be 0 for all state action pairs. Given that all SR entries are non-negative, we know that $r_{\text{SR-R}}$ only admits negative rewards, hence the zero-initialisation yields optimistic initialisation, which encourages the agent to explore [40, 29]. To disentangle the effect of SPIE from optimistic initialisation, we perform the ablation study on pure exploration with augmented SARSA-SR and SARSA-FR agents with optimistic initialisation. Specifically, we note that the maximum value the SR entries can take is $\frac{1}{1-\gamma}$, and additionally since the FR entries, by definition, are always less than or equal to the corresponding

[5]see the attached videos in supplementary materials for the full exploration dynamics for the considered agents

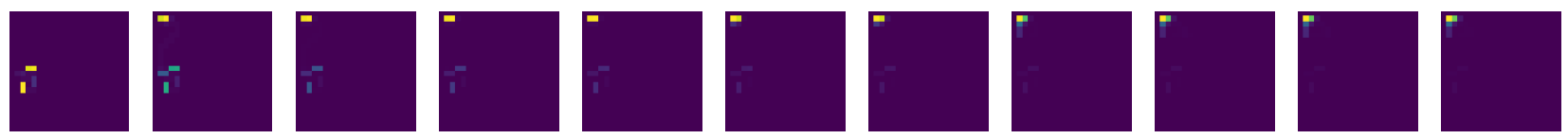

(a)

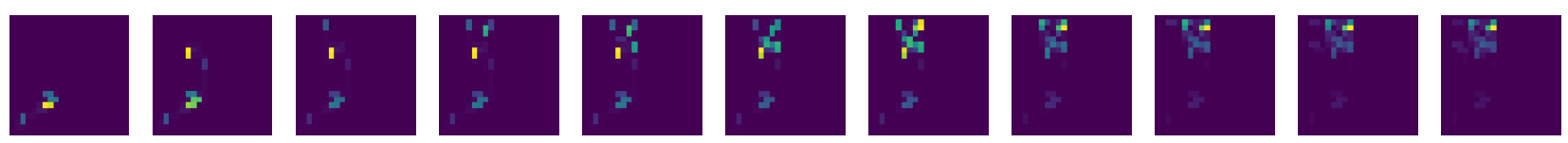

(b)

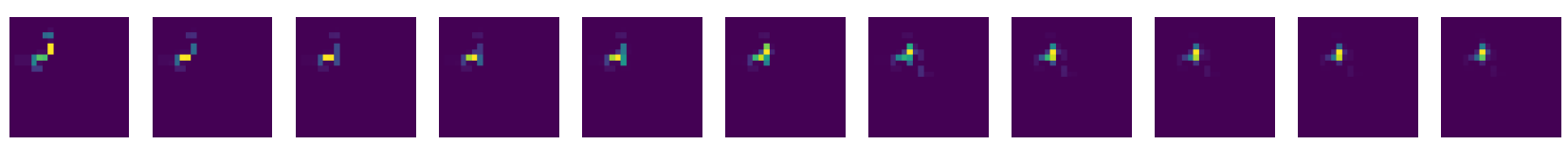

(c)

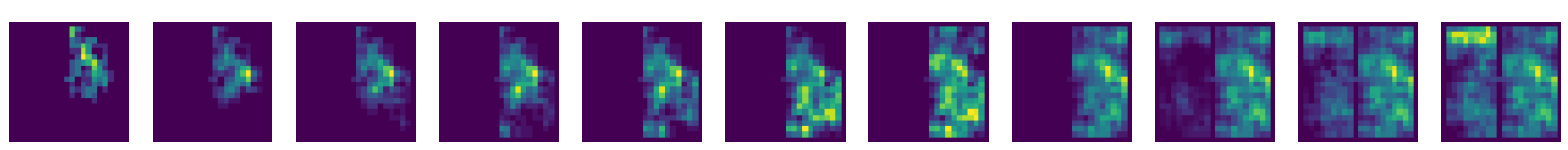

(d)

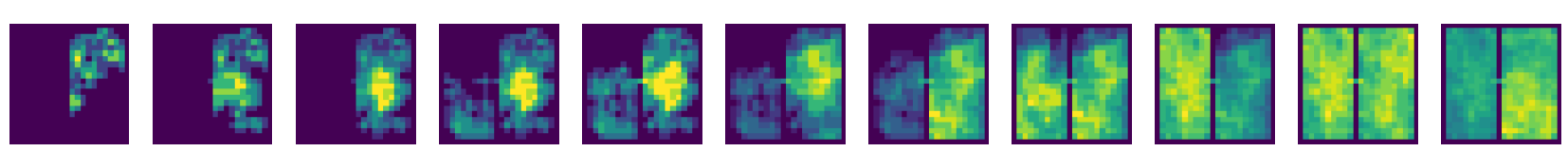

(e)

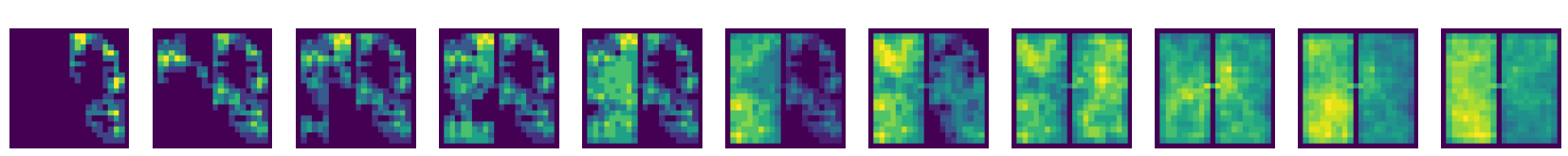

(f)

Figure 6: **Pure exploration given fixed SR / FR measures.** Temporal evoluation of state coverage heatmaps over $6000$ training steps of (a) SRASA-SR; (c) SARSA-FR; (e) SARSA-SRA agents with intrinsic rewards based on fixed SR/FR measures in *OF-small*; and (b), (d), (f) for the counterparts with online-trained SR/FR measures in the $20 \times 20$ *Cluster-simple* grid world. From left to right: $200, 400, 600, 800, 1000, 1500, 2000, 3000, 4000, 5000, 6000$ steps.

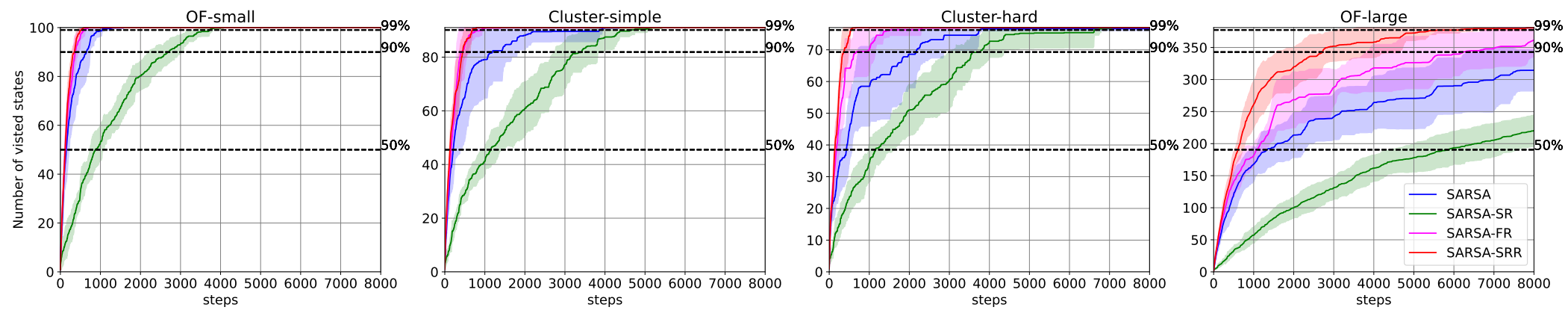

Figure 7: **Ablation study on optimistic initialisation on exploration efficiency.** We evaluate SARSA, SARSA-SRR, and optimistically augmented SARSA-SR and SARSA-FR on the considered grid worlds (Figure 1a).

SR entries, we initialise the Q values for all state-action pairs for both SARSA-SR and SARSA-FR to be $\frac{1}{1-\gamma}$. We evaluate the exploration efficiency for the optimistically augmented agents on the grid worlds (Figure 7), and we observe that despite the optimistic initialisation improves the performance of both SARSA-SR and SARSA-FR relative to their corresponding naive counterparts, the performance differences in terms of exploration efficiency between the augmented agents and SARSA-SRR are significant, hence justifying the utility of the SPIE framework independent of the optimistic initialisation.

# D Further results on deep RL implementation of SPIE in Atari games

## D.1 Ablation study on the effect of predictive reconstruction auxiliary task

In our implementation of DQN-SF-PF, by following relevant literature [27, 12], we include an additional sub-module in the neural architecture for predicting action-dependent future observation, which is trained via minimising the predictive reconstruction error. The purpose of including this sub-module is purely for learning better latent representations underlying the visual observation. We validate the utility of such predictive reconstruction auxiliary supervision by performing ablation study. We implemented an alternative version of DQN-SF-PF, removing the visual reconstruction sub-module, and test on Montezuma's Revenge. The resulting model achieves $551.5$ points (averaged over $5$ random seeds, s.e. equals $618.4$). We observe that there is a significant decrease from standard DQN-SF-PF (Table 2), indicating the importance of stronger representation learning given the predictive reconstruction auxiliary task. Moreover, given the reported performance of $398.5$ points (s.e., equals $230.1$) of DQN-SF in the absence of predictive reconstruction auxiliary task from Machado et al. [12], we observe that the SPIE objective still yields improved performance over exploration with SF alone, justifying the utility of SPIE irrespective of the specific neural architecture we choose.

# E Experiment Details

Here we provide further details of the experiments presented in the main paper.

**Tabular tasks.** We run hyperparameter sweeps for all considered agents (SARSA, SARSA-SR, SARSA-FR, SARSA-SRR) on the following hyperparameters: $\{0.005, 0.05, 0.1, 0.25, 0.5\}$ for learning rate of TD learning for the Q values ($\alpha$); $\{0.005, 0.05, 0.1, 0.25, 0.5\}$ for learning rate of TD learning for the SR/FR matrices ($\eta$); $\{0.5, 0.8, 0.9, 0.95, 0.99\}$ for the discounting factor defining the SR/FR formulation ($\gamma_{\text{SR/FR}}$); $\{1, 10, 50, 100, 1000, 10000\}$ for the multiplicative scaling factor controlling the scale of the intrinsic rewards ($\beta$); $\{0.01, 0.05, 0.1\}$ for the degree of randomness in $\epsilon$-greedy exploration ($\epsilon$). The complete sets of optimal hyperparameters for the reported performance of the considered agents in Table 1 (and for the corresponding agents with intrinsic rewards based on fixed SR/FR matrix; Table 3) in shown in Table 5.

**Exploration in grid worlds.** For all presented results in the grid worlds, we use the hyperparameters $(0.1, 0.1, 0.95, 0.95, 1.0, 0.1)$ for $(\alpha, \eta, \gamma, \gamma_{\text{SR/FR}}, \beta, \epsilon)$.

**MountainCar experiment.** We use the $128$-dimensional random Fourier features, defined over the two-dimensional state space (location$\times$speed), as the state representation. We use the hyperparameters

Table 5: Hyperparameters for the considered agents in the tabular hard-exploration tasks (the values in parentheses are the corresponding hyperparameter values for the learning of the PR).

| | agent | $\alpha$ | $\eta$ | $\gamma$ | $\gamma_{\text{SR/FR}}$ | $\beta$ | $\epsilon$ |
|---|---|---|---|---|---|---|---|
| **RiverSwim** | SARSA | 0.005 | - | 0.95 | - | - | 0.01 |
| | SARSA-SR | 0.25 | 0.1 | 0.95 | 0.95 | 10 | 0.1 |
| | SARSA-FR | 0.25 | 0.01 | 0.95 | 0.95 | 50 | 0.1 |
| | SARSA-SRR | 0.1 | 0.25 | 0.95 | 0.95 | 10 | 0.01 |
| | SARSA-SR-PR | 0.25 | 0.25(0.1) | 0.95 | 0.95(0.99) | 1 | 0.01 |
| | SARSA-SR (fixed) | 0.01 | - | 0.95 | 0.95 | 10 | 0.05 |
| | SARSA-FR (fixed) | 0.1 | - | 0.95 | 0.95 | 10 | 0.1 |
| | SARSA-SRR (fixed) | 0.25 | - | 0.95 | 0.95 | 10 | 0.01 |
| **SixArms** | SARSA | 0.5 | - | 0.95 | - | - | 0.01 |
| | SARSA-SR | 0.1 | 0.01 | 0.95 | 0.99 | 100 | 0.01 |
| | SARSA-FR | 0.1 | 0.01 | 0.95 | 0.99 | 100 | 0.01 |
| | SARSA-SRR | 0.01 | 0.01 | 0.95 | 0.99 | 10000 | 0.01 |
| | SARSA-SR-PR | 0.05 | 0.25(0.25) | 0.95 | 0.95(0.99) | 10 | 0.01 |
| | SARSA-SR (fixed) | 0.5 | - | 0.95 | 0.95 | 1 | 0.01 |
| | SARSA-FR (fixed) | 0.5 | - | 0.95 | 0.95 | 1 | 0.01 |
| | SARSA-SRR (fixed) | 0.5 | - | 0.95 | 0.95 | 10 | 0.01 |

$(0.1, 0.2, 0.2, 0.99, 0.95, 0.95, 1000, 0.3)$ for $(\alpha, \eta, \eta_{\text{PR}}, \gamma, \gamma_{\text{SR}}, \gamma_{\text{PR}}, \beta, \epsilon)$, where $\eta_{\text{PR}}$ and $\gamma_{\text{PR}}$ are the learning rate and discounting factor values for the PR, respectively.

**Atari experiments.** The neural architecture of the deep RL implementation shown in Figure 2, here we provide the specific hyperparameters of the architecture. The *Conv* block is a convolutional network with the configuration $(4, 84, 84, 0, 2) - ReLU - (64, 40, 40, 2, 2) - ReLU - (64, 6, 6, 2, 2) - ReLU - (64, 10, 10, 0, 0) - FC(1024)$, where the tuple represents a 2-dimensional convolutional layer with the architecture (num_filters, kernel_width, kernel_height, padding_size, stride), and $FC(1024)$ represents a fully connected layer with 1024 hidden units. We take the output of the *Conv* block as the 1024-dimensional state representation given the observation, which is then subsequently used for computing the SF and the PF. The action input is transformed into a high-dimensional embedding through a linear transformation, $FC(2048)$. The MLP for the predictive reconstruction block is $FC(2048) - ReLU$, for the Q-value estimation block is $FC(|\mathcal{A}|)$, for the SF head block is $FC(2048) - ReLU - FC(1024)$, for the PF head block is $FC(2048) - ReLU - FC(1024)$. The *Deconv* block is $FC(2048) - FC(1024) - ReLU - FC(6400) - Reshape((64, 10, 10)) - \langle 64, 6, 6, 2, 2 \rangle - \langle 64, 6, 6, 2, 2 \rangle - \langle 1, 6, 6, 0, 2 \rangle - Flatten$, where the tuple represents a 2-dimensional deconvolutional layer with parameters $\langle$ num_filters, kernel_width, kernel_height, padding_size, stride $\rangle$.