# OpenReview forum: "Successor-Predecessor Intrinsic Exploration"
_NeurIPS.cc/2023/Conference — NeurIPS 2023 poster_

### Official Review · Reviewer_7ZrB · 2023-06-27

**Soundness:** 2 fair
**Presentation:** 2 fair
**Contribution:** 2 fair
**Rating:** 3
**Confidence:** 4

**Summary:**

This paper is about exploration in Reinforcement Learning. Many existing works do not leverage retrospective information when computing intrinsic rewards. To this end, the paper introduces a method called SPIE where the intrinsic reward is calculated by combining successor and predecessor representations to account for prospective and retrospective information, respectively. There are different variants for discrete state space and continuous state space respectively. For discrete state space, the paper presents results on grid worlds of varying difficulty to test the efficacy of agents at covering the state space and adaptation when rewarding states change (in a continual setting). For continuous state space, comparisons on hard-exploration tasks in the Atari benchmark are presented where gains in performance are observed on a few tasks.

**Strengths:**

1. Exploration with SR has been studied in prior works. The idea of incorporating PR information in the intrinsic reward is interesting.
2. The experiments for discrete state space are thorough and show that the gains are coming from adding retrospective information with both fixed/learned SR.


**Weaknesses:**

1. The motivation is not very clear. The writing switches between learning continual exploration policy or exploration for singleton tasks. More comments on this are in Questions below.
2. The paper misses some citations and baselines. There should be discussions on learning exploratory policies that can continually explore and more recent baselines on the same. Comparisons should also be done on environments for non-stationary continuous state space environments.
3. The results on the Atari benchmark are not very convincing and more insights/tasks are needed for evaluation.


**Questions:**

1. There are methods focusing on using the retrospective structure of the transition sequences- NovelD [1]. Furthermore some recent methods have also relied on episodic novelty in addition to global intrinsic reward to improve exploration- NGU [2], E3B [3]. How does the proposed method compare with them?

2. Between lines 29-33, it is mentioned that the existing method (as they don't use retrospective information) may fail to visit bottleneck states. It is not clear why this will happen. Can this be shown experimentally on the grid world example? Also, it is possible that the intrinsic reward for visiting the bottleneck state can be small. Still, the agent will receive high rewards on going to the other side of the cluster, resulting in a higher Q-value and encouraging the agent to visit those states.

3. Lines 130-134 talk about continual exploration but focus on single-task settings. The NovelD [1] paper talks about the Asymptotic Inconsistency (page 4) of the exploration bonus so the agent can learn to exploit and converge to an optimal policy for the given task. For singleton tasks why having continual exploration is important? Furthermore, the paper does not discuss methods focusing on continual exploration or compare with them.

4. Lines 153-159 show that the proposed agent has cyclic behavior and mentions that this is advantageous for a non-stationary reward structure. However, the experiments are mostly designed for singleton tasks. Several methods have studied adaptation in continual tasks [4] or procedurally generated tasks [5,6,7]. How does the proposed method compare with them?


5. Was the proposed method also evaluated on how much coverage the agent can achieve in a single episode of finite length to validate if the method is suitable for non-stationary tasks?


6. Line 126 mentions that SR-based exploration is not ideal as the asymptotic behavior is uniform across states due to the norm becoming a fixed constant. Won’t the reward based on SF+PF face the same issues as the norm will be constant upon convergence? This would not help with continual exploration as discussed for the discrete case and may lead to different behaviors than the reward defined for the discrete state space scenario.

7. Experiments on Atari: There is no reasoning provided on why the method does better on Montezuma’s Revenge and Private Eye, and does poorly on Freeway and Venture. [5] also compared on Gravitar and Solaris, it would be interesting to see comparisons on these 2 environments too. Furthermore, more recent works should be included in the baselines like SPR [8].

8. Minor typo: Line 127: were -> where

### References:
[1] Zhang, Tianjun, et al. "Noveld: A simple yet effective exploration criterion." Advances in Neural Information Processing Systems 34 (2021): 25217-25230.\
[2] Badia, Adrià Puigdomènech, et al. "Never give up: Learning directed exploration strategies." arXiv preprint arXiv:2002.06038 (2020).
[3] Henaff, Mikael, et al. "Exploration via Elliptical Episodic Bonuses." arXiv preprint arXiv:2210.05805 (2022).\
[4] Lehnert, Lucas, Stefanie Tellex, and Michael L. Littman. "Advantages and limitations of using successor features for transfer in reinforcement learning." arXiv preprint arXiv:1708.00102 (2017).\
[5] Zha, Daochen, et al. "Rank the episodes: A simple approach for exploration in procedurally-generated environments." arXiv preprint arXiv:2101.08152 (2021).\
[6] Raileanu, Roberta, and Tim Rocktäschel. "Ride: Rewarding impact-driven exploration for procedurally-generated environments." arXiv preprint arXiv:2002.12292 (2020).\
[7] Wang, Kaixin, et al. "Revisiting intrinsic reward for exploration in procedurally generated environments." The Eleventh International Conference on Learning Representations. 2023.\
[8] Schwarzer, Max, et al. "Data-efficient reinforcement learning with self-predictive representations." arXiv preprint arXiv:2007.05929 (2020).


**Limitations:**

Some limitations are discussed in the paper. The discussion can include some potential directions on extending the method to non-stationary tasks.

---

> ### Author Rebuttal · Authors · 2023-08-07
>
> We thank the reviewers for their instructive comments. Please find below our replies to the raised concerns/questions.
>
> - We thank the reviewers for pointing us to the related literature on constructing intrinsic reward based on episodic information and retrospective transitions. SPIE is similar to NovelD in that both methods constructs the intrinsic reward based on the difference of novelty measures at the next state and the current state. However, the difference between SPIE and all three mentioned methods is that the SPIE intrinsic reward is based solely on a global intrinsic bonus, in contrast to the three methods which all leverage (partially) episodic intrinsic bonus. We will include these discussions on the relationship and differences with these methods and included citations to the mentioned papers in the related works section in the camera-ready revision upon acceptance.
>
>
> - Existing intrinsic exploration methods can nevertheless visit bottleneck states. However, consider the uncertainty based methods (e.g., RND and ICM), upon multiple visitation to the bottleneck states, the associated novelty (uncertainty) measure decreases and the agent would generate an intrinsic motivation against visiting the state in the long run. In contrast, the fundamental principle of SPIE is the dynamic balancing between the motivations of visiting bottleneck states and exploring uncertain state. Hence, the implicit motivation towards bottleneck states for making sensible exploratory transitions is sustained irrespective of the agent's perceived knowledge (uncertainty) of the bottleneck states.
>
> - Continual exploration with SPIE is potentially useful even when the task structure (e.g., its connectivity pattern) does not change. For instance, consider an ethologically plausible setting where an animal is performing random foraging in a clustered environment, where the rewards are food that is depleted upon consumption, and new rewards would occur in a different location (with a potential temporal lag). In this case, continual exploration is necessary for fast adaptation to the non-stationary reward location (as demonstrated in Figure 3c-d). However, we note that such "cycling" exploratory behaviour and the associated ethological plausibility arises as one of the benefits of SPIE, but is not our main focus in the current paper. We hence do not further study its utility under the continual exploration setting here but it is indeed an interesting direction for future investigation which we will pursue.
>
> - Similar to our argument in the previous point, the continual exploration is not the main focus of the paper. Figure 3c-d and associated discussions aims to demonstrate that SPIE exhibits the potential of showing ethologically plausible continual exploration. We do not interpret SPIE as a state-of-the-art method for solving the continual RL task or procedurally generated tasks, hence we do not provide the corresponding empirical evaluations in the paper. However, we do think this is an important and interesting direction to pursue.
>
> - We thank the reviewer for the suggestion. The average size of the set of unique states visited in a finite-length episode is reported in the rebuttal letter.
>
> - We note that the argument with respect to the asymptotic uniformity only applies to discrete setting. As the reviewer correctly points out, the SF and PF would converge asymptotically and does not lead to continual exploration. However, neither our motivation nor the focus of our empirical evaluation is focused on promoting SPIE as a method for continual exploration or examining its utility in that respect. In all our empirical evaluations under continuous state space, we only focus on single-task settings, hence the asymptotic convergence is beneficial per the asymptotic inconsistency argument from the NovelD paper.
>
> - We argue that DQN-SF-PF does not perform poorly on Freeway and Venture, but it performs comparably with the strongest baselines in these two games. It is hard to probe the exact reason underlying the performance of the agents in Atari games, but we can provide intuitive explanations. One possible reason is that Montezuma's revenge and Private Eye contains more key transition states (bottleneck states) between different "rooms" hence the "bottleneck-seeking" exploratory behaviour is beneficial in such games.
> We have now included the evaluation results of DQN-SF-PF (and other baselines) on two more hard-exploration atari games, Gravitar and Solaris. Empirical results indicate that DQN-SF-PF outperforms DQN-SR on both tasks.
> We do not think SPR is a good baseline to compare against since its motivation does not stem from designing stronger exploration agent, but rather aims at learning better state representation for efficient RL. Moreover, SPR contains more complicated neural architecture and representation learning module than DQN-SF-PF, which leads to unfair comparison. The main aim of our experimental studies is to demonstrate that SPIE is an easy-to-plugin exploration strategy that could yield the same agent to achieve more efficient exploration, rather than showing it beats all state-of-the-art models across all benchmarks.
>
> - In line 127, by "were the SR matrix to be known...", we mean that "assuming the SR matrix is known...". We do not think it is a typo, however we do apologise for the confusion.
>
> - We thank the reviewer for pointing out additional points for discussing the limits and future works of SPIE, which we will include in the discussion chapter in the camera-ready revision upon acceptance.
>
> We wish to thank the reviewer again for their thoughtful comments, and we hope the above replies address all concerns/questions raised by the reviewer. If so, we hope the reviewer could adjust their score accordingly. If there is any remaining confusion/concern, please let us know as soon as possible and we are happy to engage in further discussion.

---

> > ### Comment · Reviewer_7ZrB · 2023-08-12
> >
> > Thank you for the detailed response.
> >
> > 1) The authors clarified that continual exploration is just an application, but in the written paper it has been used for motivating the proposed method a couple of times. I do feel there should be experiments on this as there are claims made in the paper (Introduction and Motivation).
> > 2) The results on the number of states visited show that the proposed method does better. However, it is unclear how many states are there in the environment. The plot should of the number of visited states as a fraction of the total states which will show how far the algorithm is from optimal performance.
> > 3) The discussion in general response on "reciprocal form of SPIE objective in continuous settings." is important and should be in the paper. Also, I feel there should be an ablation to show the gap in performance. As for discrete state space scenarios. the method was motivated by the drawbacks of using reciprocal form.
> > 4) Regarding the response on bottleneck states, is it possible to show it with an experiment?
> >
> > Overall, the paper needs more work in terms of writing and motivating the ideas, and experiments to support the points made in the Introduction and Methods section.

---

> > > ### Author Response · Authors · 2023-08-14
> > > **Thank you for engaging in the further discussion and for the additional questions.**
> > >
> > > We thank the reviewer for reading through our responses and for their additional comments. Below we address the additional questions.
> > >
> > > - As we have stated in the responses to the reviewer and in the general response, the continual exploration is a side benefit of the model which we found interesting, but is not the main focus of the current paper. We apologise for the confusion, and we will make sure to update the paper such that the motivation is more coherent with the experiments in the paper (which we have already done but cannot show to the reviewer at the current instance since we are not allowed to upload a revised version of the paper during rebuttal this year). Here we hope to emphasise again that the main focus of the paper is to propose SPIE as an intrinsic exploration framework for more efficient exploration, but not for continual exploration (which is a surprising side benefits that we wish to pursue in future works).
> > >
> > > - There are 100 states, 91 states, 77 states, 400 states in OF-small, Cluster-simple, Cluster-hard, and OF-large environments, respectively. We set the maximum episode lengths to be 200 across all the environments. We hence observe that SARSA-SRR achieves near-optimal exploration performance across the 4 environments. We will reformat the figures by plotting the number of unique visited states as a fraction of the total number of states in the camera-ready revision upon acceptance.
> > >
> > > - We will include the discussion on the choice of reciprocal form of SPIE objective in continue settings in the camera-ready revision upon acceptance. We thank the reviewer for suggesting the additional ablation study on verifying the SPIE objective. However, given the large computational costs for hyperparameter tuning for the alternative model (difference in norms between the SF and PF), we are unable to deliver the ablation results at the current instance. We will include this ablation study in the updated version of the paper.
> > >
> > > - Yes we can show the directed intrinsic motivation towards the bottleneck states. We have videos showing the dynamics of the agent with associated value functions which we will upload as part of the supplementary materials in the updated version. In the meantime, we have created an official comment for AC containing an anonymised link to this video (but we are afraid that we cannot share it here given the regulations this year). Moreover, as suggested by Reviewer dj2L, we will also show the value function / intrinsic reward heatmaps at different checkpoints through learning to justify the dynamic balancing between exploring bottleneck states and less visited states.
> > >
> > > We thank the reviewer again for engaging in further discussion and for the additional questions. We hope the reviewer for adjust their score accordingly if the above responses have clarified the additional questions raised by the reviewer. If there is any remaining confusion/concern, please let us know as soon as possible and we are happy to engage in further discussion.

---

### Official Review · Reviewer_dj2L · 2023-07-04

**Soundness:** 2 fair
**Presentation:** 2 fair
**Contribution:** 2 fair
**Rating:** 5
**Confidence:** 3

**Summary:**

This paper introduces a new intrinsic reward (termed SPIE) for encouraging exploration in reinforcement learning. Unlike existing methods that mainly focus on prospective information, SPIE also incorporates retrospective information into the intrinsic reward. SPIE combines successor representation (SR) and predecessor representation (PR). The effectiveness of SPIE is validated in experiments on discrete grid worlds, continuous MountainCar, and 4 Atari games.

**Strengths:**

- The idea of combining both prospective and retrospective information in exploration is interesting.
- This paper is in general clear and well-written.

**Weaknesses:**

 - Beyond the performance gain, there lack in-depth discussions on the disadvantages of solely using prospective information and the benefits of adding retrospective information.
 - Line 105: The authors mention that retrospective information incorporates the connectivity about the state space. I am not sure if it can be considered an advantage that is specific to retrospective information. In my understanding, the prospective SR is also able to capture the connectivity between two states.


**Questions:**

Apart from the points raised above, I have some questions and comments:
- Figure (a): Can the authors comment on why SARSA-SR performs much worse than random walk (SARSA)? Maybe it is limited to such tabular cases. Usually, random exploration is not a strong baseline in high-dimensional domains.
- Line 40: What are "some of the problems"?
- Line 195: How does it perform if we define $r_\textrm{SR-R}$ in a similar way as in discrete MDPs?
- What if we define the intrinsic reward as $r(s,a,s')=\lVert\hat{M}[s,:]\rVert_1 - \hat{M}[s,s']$? The intuition is to encourage transitions that are less reachable from $s$. How does it perform compared to SPIE?
- Table 1 is wider than the text width.
- Figure 4(c): Please include the standard deviations of the results in the figure.

**Limitations:**

The limitations of the proposed method are barely discussed.

---

> ### Author Rebuttal · Authors · 2023-08-07
>
> We thank the reviewers for their instructive comments. Please find below our replies to the raised concerns/questions.
>
> - The retrospective information can be leveraged for identifying essential predictors for reward states, hence exploration using the retrospective information allows the agent to traverse the predictor (in our case, the bottleneck states) frequently regardless of the immediate information gain. In addition to demonstrate empirical performance on benchmarks, we also conducted extensive experimental studies in grid worlds to demonstrate the utility of SPIE framework. Specifically, we show that SPIE leads to improved exploration efficiency and state coverage in the complete absence of extrinsic reward. We additionally show that SPIE yields cycling exploratory behaviour, i.e., traversing the bottleneck state to reach other clusters upon reaching sufficient coverage of the current cluster, enabling the agent to adapt well to non-stationary environment. We show that the utility of the discrete SPIE intrinsic reward, $r_{\text{SR-R}}$ does not arises solely from its negative nature (hence optimistic initialisation). We also show that the advantage of SPIE lies in its formulation of combining the prospective and retrospective information, by demonstrating that $r_{\text{SR-R}}$ with fixed SR also yields strong exploration efficiency. If there is any additional analysis the reviewer thinks would be useful for demonstrating the utilities of SPIE, we are happy to conduct such analysis and engage in further discussion.
>
> - The reviewer is correct that the prospective information (i.e., the SR) also contains the connectivity information. However, the retrospective information provides a marginal accessibility of the target state (s') from all other states, hence providing a better sense of the local connectivity pattern of the target state that could be useful for, e.g., distinguishing states in the centre of the environment and bottleneck states.
>
> - One possible explanation for the poor performance of SARSA-SR is its equivalence to count-based exploration (Machado et al., 2018). Count-based exploration algorithms perform poorly in larger state space with sparse reward settings, such as grid worlds. Such agents preferentially explore local region for an extended time before moving to novel sub-region, hence leading to poor exploration efficiency.
>
> - Existing intrinsic exploration methods exhibit a number of problems. For instance, a notable problem is undirected exploration, whereas SPIE provides explicit intrinsic motivation towards bottleneck states. Additionally, as the agent continues exploring the state space, the intrinsic bonus decreases to 0 asymptotically, hence does not provide any intrinsic bonus for facilitating sustained exploration, which would lead to slow response to non-stationarity. Another issue associated with existing intrinsic exploration methods is the non-stationary reward structure imposed by the additive intrinsic bonus. Here we show that SPIE yields a sensible exploration bonus even with fixed SR (PR), hence resolving the non-stationary intrinsic reward issue whilst maintaining efficient exploration.
>
> - We find that defining the SPIE objective in continuous state space as the difference in norms of SF and PF leads to difficult hyperparameter tuning and we do not have the computational resources to run comprehensive grid. The reciprocal form yields robust performance on a range of $\beta$ values (0.03-0.06). We hence choose to instantiate DQN-SF-PF with equation 16.
>
> - Using $r(s, a, s') = ||\hat{M}[s, "]||_{1} - \hat{M}[s, s']$ yields intrinsic exploration based solely on state uncertainty, but does not provide any targeted exploration motivation. However, a key principle underlying SPIE is dynamically balancing exploring under uncertainty and the implicit motivation towards bottleneck states. We implemented the agent with the suggested intrinsic bonus on both Riverswim and SixArms. The empirical performance of the resulting agent on the two tasks are $95,961\pm181,216$ and $562,346\pm 1,749,455$, respectively, which is significantly lower than the performance of SARSA-SRR on the two tasks ($2,547,156\pm 479,655$ and $2,199,291\pm1,024,726$, respectively).
>
> - We thank the reviewer for pointing out the formatting issue, we have now reduced the width of Table 1.
>
> - We thank the reviewer for pointing out the missing of standard error in Figure 4c, which we now include in the rebuttal letter. Given some further hyperparameter tuning, the resulting agent with $r_{\text{SF-PF}}$ now yields a significantly higher average completion probability than the agent with $r_{\text{SF}}$ comparing to Figure 4c in the current main paper, further justifying the utility of SPIE.
>
> - The limitations of SPIE mainly lie in its theoretical underpinning. In the paper we empirical verified a number of benefits brought by SPIE. However, it is hard to analytically derive the asymptotic properties of SPIE. For instance, is there a fixed point to the non-Markovian reward decision process problem in the complete absence of extrinsic reward? Further work is required in this direction. Moreover, our definition of SPIE objective in the discrete setting coincides with the successor contingency definition [Gallistel et al., 2014], and the SC quantity is shown to contain useful information about the essential predictor of reward states. Our analysis in the current paper does not arise from a causal perspective, which is a promising venue for future investigation.
>
> We wish to thank the reviewer again for their thoughtful comments, and we hope the above replies address all concerns/questions raised by the reviewer. If so, we hope the reviewer could adjust their score accordingly. If there is any remaining confusion/concern, please let us know as soon as possible and we are happy to engage in further discussion.

---

> > ### Comment · Reviewer_dj2L · 2023-08-12
> > **Feedback**
> >
> > Thank the authors for the response. I still have some questions and would appreciate the authors' feedback:
> > - Can the authors visualize the intrinsic reward heatmap for environments in Fig1? For experiments in Fig1(b), as the SR is learned online, the authors can plot intrinsic reward heatmaps at several checkpoints. For experiments in Fig1(c), a single plot should suffice as SR is fixed.
> > - In Line 128, the authors mention that exploring with $r_\text{SR}$ would regress back to random exploration if the SR matrix were known a priori. Then why does SARSA-SR perform worse than random exploration in Fig1(c)?

---

> > > ### Author Response · Authors · 2023-08-14
> > > **Thank you for engaging in the further discussion and for the additional questions.**
> > >
> > > We thank the reviewer for reading through our responses and for their additional comments. Below we address the additional questions.
> > >
> > > - We have visualised the intrinsic reward as heatmaps for different models. We are afraid that we are unable to upload the new visualisation at the current stage of the rebuttal, hence we will verbally describe the the intrinsic reward heatmaps. We describe the dynamics of the intrinsic reward heatmaps for the 20x20 Cluster-simple environment.
> > >
> > > For the online learning case, at the early phase of exploration when the agent initially explores the environment, the SR matrix is updated over the visited states, hence the intrinsic rewards becomes negative over the states visited and remains zero for unvisited states, facilitating exploration towards unvisited states. As the learning continues, i.e., upon sufficient coverage of the environment and learning of the SR matrix, the resulting intrinsic rewards exhibit dynamic balancing between moving towards the bottleneck states and exploring remaining states in the current cluster. The dynamic balancing depends on the recent trajectory taken by the agent and the amount of time the agent spends in the current cluster since last visitation. As the agent spends more time in the current cluster, the intrinsic rewards towards the bottleneck states become greater.
> > >
> > > For the fixed-SR case, the intrinsic rewards are stationary over the training process. We observe that the intrinsic rewards towards the direction to the bottleneck state are higher for the states closer to the bottleneck state, and is more evenly spread out over the action space for the states towards the centre of each cluster.
> > >
> > > The intrinsic reward heatmaps indicate that SPIE induces a generic intrinsic motivation towards the bottleneck states, but is balanced by the motivation towards states of high uncertainty (less visited states), hence yielding more efficient exploration. We apologise for not being able to show the plots right now, but we will include the heatmaps in the camera-ready revision upon acceptance.
> > >
> > > - When the SR matrix is known as fixed a priori, SARSA-SR would regress back to random exploration only upon reaching converged value function. Throughout our analysis, we have assumed the Q-values are initialised to 0 for all state-action pairs. Therefore, the first action (regardless of the action taken) yields a positive reward (1/20 when $gamma = 0.95$), leading to positive Q-value for the corresponding state-action pair. This process goes on and the agent would follow the same path repeatedly by acting greedily with respect to the Q-values, with the only exceptions at the states where the agent has visited more than once over a single trajectory (due to randomness of the Q-value initialisation or epsilon-greedy). Asymptotically, the action value function converges and the behaviour of the SARSA-SR agent is identical to a purely random exploration agent.
> > >
> > > We thank the reviewer again for engaging in further discussion and for the additional questions. We hope the reviewer for adjust their score accordingly if the above responses have clarified the additional questions raised by the reviewer. If there is any remaining confusion/concern, please let us know as soon as possible and we are happy to engage in further discussion.

---

> > > > ### Comment · Reviewer_dj2L · 2023-08-15
> > > > **Thank you for the reply**
> > > >
> > > > Thank the authors for the updates.
> > > >
> > > > The descriptions of the visualizations make sense. It would be better if the authors can add anonymous links to those visualizations.
> > > >
> > > > In the fixed SR setting (Fig1c), if I understand it correctly, the intrinsic reward is the same for every state. I am wondering if the performance of SARSA-SR might be related to the scale/sign of the reward. [1] observes that a positive reward shifting leads to conservative exploitation, while a negative reward shifting leads to curiosity-driven exploration. Among the methods in comparison, SARSA-SRR is the only one with a negative reward. I would appreciate it if the authors can run SARSA experiments with different constant intrinsic rewards $r$. For example, $r$ takes values from a range $[-\alpha, \alpha]$ where $\alpha$ can be the average $\lvert r_\text{SR-R}\rvert$ across all states.
> > > >
> > > > [1] Sun, Hao, et al. "Exploit Reward Shifting in Value-Based Deep-RL: Optimistic Curiosity-Based Exploration and Conservative Exploitation via Linear Reward Shaping." Advances in Neural Information Processing Systems 35 (2022): 37719-37734.

---

> > > > > ### Author Response · Authors · 2023-08-15
> > > > > **Thank you for engaging in further discussion**
> > > > >
> > > > > We thank the reviewer for engaging in further discussion.
> > > > >
> > > > > We are afraid that we are unable to add anonymous links to the visualisations per the regulations this year ["All the texts you post (rebuttal, discussion and PDF) should not contain any links to external pages."...]. However, we will include such visualisation in the camera-ready revision upon acceptance.
> > > > >
> > > > > We thank the reviewer for pointing out the possible confounding factor of reward scales. We recognised the fact that SARSA-SRR yields negative rewards, hence the improved exploration might be attributed to the confounder of optimistic initialisation. We performed ablation studies by examining the performance of SARSA-SR with Q-values initialised to be the maximally attainable values ($\frac{1}{\gamma}$). Please see Appendix Figure 7 and corresponding discussions for more details. As suggested by the reviewer, we normalise and rescale the intrinsic reward of SARSA-SR such that the $r_{\text{SR}}$ now takes values in the range of $[-\alpha, \alpha]$. By implementing the resulting agent in the grid worlds under pure exploration settings (Figure 1), the performance improves but is still significantly lower than SARSA-SRR. We will include these new ablations in the updated paper.
> > > > >
> > > > > We again thank the reviewer for engaging in extended discussion. We are happy to engage in further discussions if there is any remaining question from the reviewer.

---

> > > > > > ### Comment · Reviewer_dj2L · 2023-08-15
> > > > > > **Feedback**
> > > > > >
> > > > > > Thank the authors for participating in the discussions. Maybe I did not explain it clearly. I was wondering if the poor exploration performance of SARSA-SR in Fig1c is due to positive rewards. What I was suggesting in my previous comment is SARSA experiment with **constant** intrinsic rewards. I do not think it has anything to do with normalizing the intrinsic reward of SARSA-SR, since in the fixed SR setting the intrinsic reward is the same across states. To give a very concrete example, the experiment I am thinking of is: running SARSA with fixed constant reward $r=-0.5$, $r=-0.3$, $r=-0.1$, $r=0.1$, $r=0.3$, $r=0.5$. The numbers here are mainly for the sake of example, please use equally-spaced values from the range $[-\alpha, \alpha]$. And when $r=0$, it should reduce to the vanilla SARSA.

---

> > > > > > > ### Author Response · Authors · 2023-08-16
> > > > > > > **Re. constant intrinsic reward**
> > > > > > >
> > > > > > > We thank the reviewer for following up on the previous questions.
> > > > > > >
> > > > > > > We apologise for misunderstanding your previous question. Theoretically speaking, the magnitude of the constant intrinsic reward is should not lead to any difference in performance in the complete absence of extrinsic reinforcement (i.e., r = 0.5, 0.3, or 0.1 should yield same/similar results). However, the sign of the constant intrinsic reward should matter. Since by setting the intrinsic rewards to negative, and with zero initialisation for the Q-values, the resulting agent becomes an instance of optimistic initilisation, which has been shown to improve exploration (see, e.g., Chosen et al., 2018 and Ciosek et al., 2019). Indeed from simulations (under the same experimental settings as in Figure 1), we observe that the resulting agents with $r_{\text{intrinsic}} \in \{0.1, 0.3, 0.5\}$ yields similar performance as SARSA-SR (with fixed SR), and $r_{\text{intrinsic}}\in\{-0.1, -0.3, -0.5\}$ yields significantly higher performance than SARSA-SR, leading to better exploration efficiency than SARSA (purely random exploration), whilst still achieving lower efficiency comparing to SARSA-SRR. Both theoretical reasoning and empirical evidence indicate that the scale of the (constant) intrinsic reward should not affect the exploration behaviour of the agent (under pure exploration settings, i.e., without external reinforcement), whereas the sign of the reward (and the existence of optimistic initialisation) would lead to significantly different behaviours and performance.  The confounding factor of optimistic initialisation has been addressed both in the original paper (appendix Figure 7) and in the above new experiments suggested by the reviewer, i.e., SPIE yields improved exploration efficiency regardless of optimistic initialisation. We will include these new ablations in the camera-ready revision upon acceptance.
> > > > > > >
> > > > > > > We hope we have addressed the question raised by the reviewer. Please let us know if you have any additional question/concern.
> > > > > > >
> > > > > > > References:
> > > > > > >
> > > > > > > [1] Choshen, L., Fox, L. and Loewenstein, Y., 2018. Dora the explorer: Directed outreaching reinforcement action-selection. arXiv preprint arXiv:1804.04012.
> > > > > > >
> > > > > > > [2] Ciosek, K., Vuong, Q., Loftin, R. and Hofmann, K., 2019. Better exploration with optimistic actor critic. Advances in Neural Information Processing Systems, 32.

---

> > > > > > > > ### Comment · Reviewer_dj2L · 2023-08-16
> > > > > > > > **Thank you**
> > > > > > > >
> > > > > > > > I sincerely appreciate the authors for their thorough responses during our discussions. I no longer have questions and my score has improved. I encourage the authors to revise the submission carefully, integrating the suggested changes.

---

> > > > > > > > > ### Author Response · Authors · 2023-08-18
> > > > > > > > > **Thank you for your response and for raising the score**
> > > > > > > > >
> > > > > > > > > We thank the reviewer for their active engagement in discussion during the rebuttal period. We are happy to see that we have addressed all questions raised by the reviewer, and we thank the reviewer for increasing their score.

---

### Official Review · Reviewer_pKLV · 2023-07-05

**Soundness:** 3 good
**Presentation:** 3 good
**Contribution:** 3 good
**Rating:** 4
**Confidence:** 3

**Summary:**

This paper proposes Successor-Predecessor Intrinsic Exploration (SPIE), an exploration framework that uses the successor representation (SR) and predecessor representation (PR) to formulate intrinsic motivation for exploration. Amongst the variations, two specific design of intrinsic rewards are highlighted: SR-R and SF-PF. While the first directly use the successor and predecessor representations to calculate the intrinsic reward for tabular case, the latter utilizes successor and predecessor features for a smooth expansion to continuous state spaces. The experiments show that the highlighted reward designs display a promising  results.

**Strengths:**

The paper proposes a unique and novel approach to exploration problem in RL. Furthermore, leveraging the successor representation displays a broad future directions of application as the successor representation are taken to be helpful in the view of generalizability in RL.

**Weaknesses:**

The presentation needs some finishing touches and variables and logics are left unexplained in places. Also the experiments seem to be done in relatively small scales, whereas the the algorithm is not restricted small scales. For example, training curves of more Atari task, trained for longer period (since hard-explorations can often take more experiences), may show more clear pattern of learning.

**Questions:**

Some questions.
- For SR-R, since $\mathbf{M}[s,s'] \le \mathbf{M}[:,s']$, the intrinsic reward is likely to be negative, which can possibly discourage the agent to proceed to a longer episodic length when terminal state exists and can be discovered. Could this be a problem to applying SR-R to such tasks?
- I did not fully follow how the learned $\phi$ preserves the property $r = \phi \cdot \mathbf{w}$. Also, what are the conditions for $\phi$ and $\mu$ to be shared? (which I suspect is the case based on Figure 2, since $\mu$ is not clearly defined or described in the paper)
- The score for RND seems far too off from what I would expect based on previous works, even regarding the number of training frames. Could the authors provide more details on the RND implementation? Also the score is missing for Freeway, whereas in the text, it reports 4 tasks (line 297).

Minor remarks.
- The sentence before Eqn. 10 and the equation itself have period at the end, which should be commas. Similar punctuation errors can be found in other equations.
- The caption for Table could be more specific.
- Tables would look more nicer if they are resized, fitted to the textwidth.
- several notations are not clearly explained, e.g., $\hat q^\pi$, $\mu$

**Limitations:**

The limitations are stated in the conclusion.

---

> ### Author Rebuttal · Authors · 2023-08-07
>
> We thank the reviewer for their instructive comments. Please find below our replies to the raised concerns/questions.
>
> - Thanks for pointing out the missing of explanations of variables and logics in the current manuscript. In responding to other reviewers' comments, we have modified our manuscript for clearer demonstration and explanation. For instance, we now make clear that the main focus of the paper is on proposing stronger intrinsic exploration algorithm rather than continual exploration in non-stationary environments. We now give a more elaborate explanation on why SARSA-SR behaves poorly in pure exploration. Moreover, we provide more details on the description of the model and associated implementations. We hope these modifications have made the paper clearer and more rigorous. If there is any additional confusion remaining, could the reviewer provide specific pointers to places where we have such omission?
>
> - Through the experimental studies, instead of only showing that it yields higher performance on different tasks, we aim to maximally demonstrate the exploratory behaviours of SPIE, hence justifying our extensive evaluations on (relatively) small-scale tasks such as grid worlds. We show the final performance (at 100M training steps) instead of the training curves since we do not have the resources to run all selected baseline algorithms on all presented Atari games, hence we reporting the asymptotic performance from existing literature (e.g., from Machado et al. 2018), which prohibits us from showing the full training curve. However, we do wish to note it is common in the literature to only report the final performance instead of the full training curve in practice, therefore we do not think it would negatively affect faithful validation of the model.
>
> - As the reviewer correctly points out, the $r_{\text{SR-R}}$ intrinsic reward is negative. However, in tasks with terminal states (usually denoting reward state), we believe it is usually preferable to reach the target state as quickly as possible. SARSA-SRR agent yields stronger exploration efficiency for faster detection of the goal states, which could then facilitate further exploration and learning for finding shorted paths towards the goal states.
>
> - The successor feature generalises SR to continuous state space, and requires a notion of state representation for constructing the similar cumulative future occupancy measure. Hence the SF is defined as the expected discounted cumulative sum of future state representations. The motivation of leveraging a state representation that could represent the reward vector as a linear transformation of the representation is based on the motivation that the SF wish to yield the similar decomposition of value function into the dot product between the SR/SF and reward vector. $\mu$ and $\phi$ are the same vector as we wish to construct the SF and the PF with the same state representation. We apologise for the missing notation. We now define define the PF using the same feature vector as the SF, $\vec\xi^{\pi}(s) = \vec\mu(s_{t+1}) + \gamma\mathbb{E}\left[\vec\xi^{\pi}(s_{t})|s_{t+1}=s, a_{t}=a\right]$ in the updated manuscript.
>
> - The evaluation of RND agent on Freeway was missing since we did not have the computational resources to finish all evaluations. We have provided the RND performance on Freeway in the updated manuscript (28.2 points, $\sigma^2 = 0.2$). The main reason behind the difference in our evaluation of RND and existing evaluations reported in the literature is the reward scheme is set differently in our implementation comparing to the original implementation from Burda et al., 2018. Specifically, we report the canonical cumulative episodic reward, whereas the original implemented bin the rewards to {+1, 0, -1} by its sign, which results in larger episodic return (see, e.g., https://github.com/openai/random-network-distillation/blob/f75c0f1efa473d5109d487062fd8ed49ddce6634/atari_wrappers.py#L220). We have now included the evaluation results of DQN-SF-PF (and other baselines) on two more hard-exploration atari games, Gravitar and Solaris. Empirical results indicate that DQN-SF-PF outperforms DQN-SR on both tasks.
>
> - We thank the reviewer for pointing out the punctuation errors, which we have corrected in the updated manuscript and will be included in the camera-ready revision upon acceptance.
>
> - We have expanded the description in the table captions for clarity. We have also reduced the width of the table such that it now fits to the textwidth. These modifications will be included in the camera-ready revision upon acceptance. Thanks for the suggestions.
>
> - We thank the reviewer for pointing out the unclear notation in the paper, which we have now included the associated explanations in the updated manuscript.
>
> We wish to thank the reviewer again for their thoughtful comments, and we hope the above replies address all concerns/questions raised by the reviewer. If so, we hope the reviewer could adjust their score accordingly. If there is any remaining confusion/concern, please let us know as soon as possible and we are happy to engage in further discussion.

---

> > ### Author Response · Authors · 2023-08-14
> > **Any additional questions?**
> >
> > We thank the reviewer again for their review. As the end of the discussion period is approaching, we kindly ask the reviewer to engage in further discussion if there is any additional question/concern remaining, and we hope the reviewer could adjust their score accordingly if all raised concerns are addressed. We look forward to your further responses!

---

> > > ### Comment · Reviewer_pKLV · 2023-08-16
> > > **Thanks for the answers**
> > >
> > > Thanks to the authors for the responses. I have few more remarks and questions.
> > > - Thanks for clearing the source for the values for baselines in Table 2. As authors have pointed out, it is a common practice to reference to scores from previous literatures but I think it would be more clear if authors could include the reference to respective sources of the scores in the main text or the caption. Additionally, to further clarify the comment, the training curve would help since RND is known to perform much differently after the 1e8 frame point, which is only a fraction of true asymptotic performance of RND [1] given more training steps. Furthermore, as authors have taken the scores from Machado et al. 2018, which states that RND scores were used since since its the state-of-the-art (SOTA) at the time and mentions that at 2e9 frames, RND still achieves the SOTA performance. As of now, it is debatable if RND is still the SOTA, as there are other works along the similar line [2], which have shown better performances compared to RND. Possibly, it could be more beneficial to include the performances of those works for a better comparison.
> > > - As for the shortest path to goal states, the authors are correct if the goal state (rewarded state) is the only possible terminal states. However, if there exist non-rewarding terminal states (i.e., agent dies without earning any rewards), I suspect there would be a trade-off between the possible discounted reward and the sum of discounted penalties (i.e., the negative rewards).
> > >
> > > [1] EXPLORATION BY RANDOM NETWORK DISTILLATION
> > > [2] NovelD: A Simple yet Effective Exploration Criterion

---

> > > > ### Author Response · Authors · 2023-08-18
> > > > **Thank you for engaging in further discussion**
> > > >
> > > > We thank the reviewer for responding to our rebuttal, and for engaging in further discussions.
> > > >
> > > > - We apologise the omission of explicit discussion on using reported performance statistics in the literature where appropriate. We will make this clear in the camera-ready revision upon acceptance. We thank the reviewer for raising the point for including more recent SOTA results for comparison and for showing the training curve comparison. We will include these additional results in the updated version with more time for evaluating the new baselines on the Atari tasks. However, we wish to point out that the main goal of the empirical evaluations presented in the paper is to show that SPIE intrinsic reward could significantly improve exploration, hence the sample efficiency of learning, which could be maximally demonstrated through comparison with the base model that SPIE is implemented upon.
> > > >
> > > > - We thank the reviewer for proposing the possible extension to scenarios with multiple non-rewarding terminal states. This is an interesting question for further investigation. We have ran some preliminary analysis for investigating the trade-off between extrinsic reward and sum of intrinsic rewards, but so far we have not found clear pattern in this respect. We will continue study such scenario, and will include related discussions in the camera-ready revision upon acceptance.
> > > >
> > > > We thank the reviewer again for engaging in further discussion and for the additional questions. We hope the reviewer for adjust their score accordingly if the above responses have clarified the additional questions raised by the reviewer. If there is any remaining confusion/concern, please let us know as soon as possible and we are happy to engage in further discussion.

---

### Official Review · Reviewer_WHHE · 2023-07-06

**Soundness:** 4 excellent
**Presentation:** 2 fair
**Contribution:** 4 excellent
**Rating:** 7
**Confidence:** 4

**Summary:**

The paper proposes to use both successor feature of s and predecessor feature of s' as intrinsic reward to improve RL exploration. Specifically, in the tabular case, the proposed intrinsic reward encourages the agent to visit states that are infrequently visited from other states, using successor representation which measures visitation frequency of s' from s and from other states. Then the method replace the visitation frequency from all states to s' with the predecessor representation of s'. Finally, to extend to continuous state spaces, the method approximate successor and predecessor representations with successor and predecessor features.

**Strengths:**

1. Intrinsic reward is an important way to improve RL exploration and thus sample efficiency, and the proposed method is well motivated (to visit bottleneck states more frequently), to the best of my knowledge, is novel.
2. The experiments are well designed, ranging from tabular to gridworld to atari to show the effectiveness of each variation of the method: (1) using successor representation along, (2) replacing row sum of successor representation with predecessor representation, and (3) replacing representations with features.
3. The background of successor representations is written clearly, preparing readers well for the proposed method.

**Weaknesses:**

1. The method section is well structured, but it can be improved in the following places:

* It's clear that successor representation means the visitation frequency from s to s', and using Eq 8 follows the proposed motivation of encouraging visitation to s' that are rare from states other than s. Then, for continuous state space, the authors replace successor representation with successor feature. But why does the successor feature could represent the visitation frequency? After the replacement, does the intrinsic reward still follow the same motivation? I feel this replacement is not well explained/motivated.
* For the successor/predecessor feature, different feature columns may have different value ranges (scales). How to account for that to avoid certain feature columns dominating the column sum?
* Any possible explanation for the form of the reciprocal of norms in Eq. 16? What are other forms that you tried but underperform?
* Why is reconstruction needed? It's unclear how much the reconstruction contributes to representation learning and exploration.
* When the authors mention the "row/column sum" of some symbol, the shape of the referred symbol (including which is row and which is column) is usually not introduced and thus confusing, including but not limited to L192.
* Figure 1: why does SARSA-SR perform so badly?
* (minor) unintroduced symbols and typos: L165 "diffusion" (you can probably remove "diffusion"), Eq 14 $\mu$, L181 N's shape missing "|", Fig 2 the lines are mixed together and thus a bit hard to read.

**Questions:**

Beyond the questions above:
* What do the agent trajectories look like? Do they follow the motivation of visiting the bottleneck state?
* Eq 8 reminds me of the motivation in the "diversity is all you need" paper where different trajectories should visit different parts of the state space. Are the two papers related? If so, how? Adding this discussion would be helpful.
* Is the reconstruction necessary? An ablation study would be helpful.
* What are the confidence intervals like in Fig 4 (c)?
* Why is RND missing in Table 2 Freeway row? Any explanation for the worse performance of the proposed method on the private eye?

**Limitations:**

The only limitation is the lack of motivation of replacing successor/predecessor representation with their corresponding features for continuous state space.

---

> ### Author Rebuttal · Authors · 2023-08-07
>
> We thank the reviewer for their instructive comments. Please find below our replies to the raised concerns/questions.
>
> - The generalisation to continuous state space means that we can no longer enumerate all states for the computation and learning of the SR. Hence we adopt a natural extension of the SR to continuous setting, the successor features. The SF also exhibits the similar predictive encoding as the SR in discrete settings, hence could also represent the prospective transition information in continuous setting (similarly the retrospective information for the predecessor features). Empirically, we find the generalisation to SF work well with our models and previous models (e.g., Machado et al., 2020) using either linear or non-linear function approximation, hence justifying our choice.
>
> - In our implementation of the SF-PF-DQN, both the SF and the PF are based on the intermediate features ($\phi_{t}$ in Figure 2), which are L2-normalised for avoiding the scaling issue. This is done with F.normalize(phi, p=2, dim=-1) (see line 105 in atari/models/sr_pr_dqn.py in the provided codes).
>
> - The initial proposal was to use the difference of norms of the SF and PF as is done for the SR in discrete settings. However, we find it hard to set the scaling factor for the SPIE intrinsic reward in this case. We hence adopt the reciprocal form presented in Machado et al., 2018 (last equation on page 5), which we find to yield robust performance on a range of $\beta_{spie}$ values (0.03-0.06). We note that the fundamental principle of SPIE is the combination of prospective and retrospective information for exploration, admitting different instantiations (including those shown in equation 8 and 16).
>
> - In our implementation of DQN-SF-PF, by following relevant literature [oh et al., 2015, Machado et al., 2020], we include an additional sub-module in the neural architecture for predicting action-dependent future observation, which is trained via minimising the predictive reconstruction error. The purpose of including this sub-module is purely for learning better latent representations underlying the visual observation. We validate the utility of such predictive reconstruction auxiliary supervision by performing ablation study with an alternative version of DQN-SF-PF, removing the visual reconstruction sub-module. Testing on Montezuma's Revenge, the resulting model achieves $551.5$ points (averaged over $5$ random seeds $\sigma^2=618.4$). We observe that there is a significant decrease from standard DQN-SF-PF, indicating the importance of stronger representation learning given the predictive reconstruction auxiliary task. Moreover, given the reported performance of $398.5$ points ($\sigma^2=230.1$) of DQN-SF in the absence of predictive reconstruction auxiliary task from Machado et al., 2020, we observe that the SPIE objective still yields improved performance over exploration with SF alone, justifying the utility of SPIE irrespective of the specific neural architecture we choose.
>
> - We will include the shape of the referred row/column sum and corresponding features, and keep the notations consistent in the updated manuscript. Thanks for pointing it out.
>
> - As mentioned in the analysis in Section 3, one possible explanation for the poor performance of SARSA-SR is its equivalence to count-based exploration (Machado et al., 2018). Count-based exploration algorithms perform poorly in larger state space such as grid worlds (in contrast to 6 and 7 states for RiverSwim and SixArms) such that the agent preferentially explores local region for an extended time before moving to novel sub-region, hence leading to poor exploration efficiency.
>
> - We thank the reviewer for pointing out the typos, which we have fixed in the updated manuscript.
>
> - We have sample video containing a sampled trajectory of SARSA-SRR in the grid world. As is also discussed in the main paper, the SARSA-SRR agent tends to balance the motivations of visiting bottleneck states and exploring uncertain state, hence the typical behavior is the agent spend some time exploring the current cluster, and upon sufficient coverage it would move directly to the bottleneck state for exploring other clusters. In the two-cluster environment, we indeed observe such behavior where the agent spend the majority of its time exploring one of the clusters and transition into the bottleneck states upon reaching sufficient coverage.
>
> - The relationship between SPIE and DIAYD lies in their connections to the empowerment objective, which facilitates diverse exploration trajectories. We thank the reviewer for pointing out the connection, which we have now included associated discussions in the related work section.
>
> - Yes the reconstruction is necessary empirically, see the fourth point above.
>
> - We have now included the new Figure 4c with standard error in the rebuttal letter.
>
> - We have now included the Freeway evaluation (28.2 points with $\sigma^2=0.2$, see Table 1 in the rebuttal letter). One possible reasoning for the relatively poor performance on Private Eye across all agents apart from vanilla DQN is that the background of the observation is frequently changing (unlike other games such as Venture and Montezuma's revenge), hence imposing large intrinsic exploration signals much more often comparing to other tasks, such that the agent spends the majority of its time exploring instead of focusing on maximising the reward. This is coherent with the observation that DQN without any intrinsic exploration behaves best on Private Eye whereas other agents with intrinsic exploration behaves poorly.
>
> We wish to thank the reviewer again for their thoughtful comments, and we hope the above replies address all concerns/questions raised by the reviewer. If so, we hope the reviewer could adjust their score accordingly. If there is any remaining confusion/concern, please let us know as soon as possible and we are happy to engage in further discussion.

---

> > ### Comment · Reviewer_WHHE · 2023-08-13
> >
> > Thanks for the detailed response, especially the ablation of using reconstruction or not! I increase my score to 7 and encourage the authors to add the extra results in the next version of the paper.

---

> > > ### Author Response · Authors · 2023-08-14
> > > **Thank you for engaging in further discussion and for raising the score.**
> > >
> > > We are glad to see that our responses addressed the concerns raised by the reviewer and we thank the reviewer for increasing their score. We will make sure to include all the new analysis in the camera-ready revision upon acceptance.

---

### Author Rebuttal · Authors · 2023-08-07

We thank all reviewers for their instructive comments for making the paper clearer and more rigorous. There are a number of questions/concerns that are shared by multiple reviewers, which here we provide summarised responses below. We have also attached a rebuttal letter containing additional experiment results requested by the reviewers.

- Main motivation of the paper and the experimental studies:

The main focus of the paper is to propose the SPIE intrinsic exploration framework of combining prospective and retrospective information for more efficient and targeted exploration. Whilst we show additional side benefits that SPIE yields ethologically plausible "cycling" exploratory behaviour, it is not the main focus nor the motivation of SPIE. We provide preliminary experimental studies illustrating the existence of such "cycling" behaviour and discuss its potential application for continual exploration for dealing with non-stationary environments. However, it is by no means our intention to claim or promote SPIE as a state-of-the-art method for dealing with non-stationary continual RL problems.

Moreover, we wish to clarify that the aim of our empirical evaluations is to demonstrate the improved exploration (and hence learning) efficiency given the dynamic balancing of exploring uncertain states and transitioning into bottleneck states admitted by SPIE. Our experiments comprehensively validates the utility of SPIE framework in tasks with discrete state spaces with tabular value estimates, and in tasks with continuous state spaces using linear and non-linear (deep RL) function approximations. All presented experiments demonstrate that SPIE can be efficiently implemented, with minimal modification to the original agent, whilst yielding significantly better performance. The goal of the paper is to show that retrospective information could and should be utilised for stronger exploration. The conceptual idea can be instantiated with any existing RL agents, and produce state-of-the-art results if needed.

- The choice of reciprocal form of SPIE objective in continuous settings.

We generalise SPIE to tasks with continuous state spaces by replacing SR/PR with SF/PF. The initial proposal was to use the difference of norms of the SF and PF as is done for the SR in discrete settings. However, we find it hard to set the scaling factor for the SPIE intrinsic reward in this case. We hence adopt the reciprocal form presented in Machado et al., 2018 (last equation on page 5), which we find to yield robust performance on a range of $\beta_{spie}$ values (0.03-0.06). We note that the fundamental principle of SPIE is the combination of prospective and retrospective information for exploration, admitting different instantiations (including those shown in equation 8 and 16 in the main paper).

- Poor performance of SARSA-SR on pure exploration in grid worlds.

One possible explanation for the poor performance of SARSA-SR is its equivalence to count-based exploration (Machado et al., 2020). Count-based exploration algorithms perform poorly in larger state space such as grid worlds (in contrast to 6 and 7 states for RiverSwim and SixArms) such that the agent preferentially explores local region for an extended time before moving to novel sub-region, hence leading to poor exploration efficiency.

- More evaluations in deep RL experiments.

We have now provided evaluation results of DQN-SF-PF and baseline agents on all 6 hard-exploration Atari games (Bellemare et al., 2013), which is shown in Table 2 in the attached rebuttal letter. DQN-SF-PF outperforms baseline agents on 4 of the games and yields comparable performance on the remaining 2.

- Missing standard error in linear RL experiment.

We thank the reviewers for pointing out the missing of standard errors in the linear RL experiment (Figure 4c). We have now provided the full evaluations with standard errors in Figure 1a in the rebuttal letter. We note that the reported results for $r_{\text{SF-PF}}$ in this case is from a different set of hyperparameters. We performed a comprehensive hyperparameter search, and the resulting agent with $r_{\text{SF-PF}}$ now yields a significantly higher average completion probability than the agent with $r_{\text{SF}}$ comparing to Figure 4c in the current main paper, further justifying the utility of SPIE.

- Typos and formatting issues.

We thank the reviewers for pointing out various typos, and specifically the width issue with Table 1 in the current main paper. We have now corrected all raised typos and fixed the formatting issues.

We wish to thank all the reviewers again for their thoughtful comments, and we hope the above replies address all concerns/questions raised by the reviewers. If so, we hope the reviewers could adjust their score accordingly. If there is any remaining confusion/concern, please let us know as soon as possible and we are happy to engage in further discussion.

---

### Decision · Program_Chairs · 2023-09-21

**Decision:**

Accept (poster)

**Comment:**

This paper has received very mixed reviews. While some reviewers appreciated the contribution of this work, others have raised some doubts about its motivation and empirical evaluation. The rebuttal seems to have cleared remaining doubts about the experiments, while the main motivation of the paper still seems to be a problem for a reviewer. However, I think that a lack of clarity in the description of the motivation is not a strong reason to reject this paper, and considering the scores, comments, and discussion phase, I tend to accept this paper.

I encourage the authors to address all the comments and to incorporate the recommended improvements in the final version.